# DCA: Graph-Guided Deep Embedding Clustering for Brain Atlases

**Mo Wang**
SUSTech & University of Warwick

**Kaining Peng**
SUSTech

**Jingsheng Tang**
SUSTech

**Hongkai Wen**
University of Warwick
hongkai.wen@warwick.ac.uk

**Quanying Liu**
SUSTech
liuqy@sustech.edu.cn

## Abstract

Brain atlases are essential for reducing the dimensionality of neuroimaging data and enabling interpretable analysis. However, most existing atlases are predefined, group-level templates with limited flexibility and resolution. We present Deep Cluster Atlas (DCA), a graph-guided deep embedding clustering framework for generating individualized, voxel-wise brain parcellations. DCA combines a pretrained voxel-level fMRI autoencoder with spatially regularized deep clustering to produce functionally coherent and spatially contiguous regions. Our method supports flexible control over resolution and anatomical scope, and generalizes to arbitrary brain structures. We further introduce a standardized benchmarking platform for atlas evaluation, using multiple large-scale fMRI datasets. Across multiple datasets and scales, DCA outperforms state-of-the-art atlases, improving functional homogeneity by 98.8% and silhouette coefficient by 29%, and achieves superior performance in downstream tasks. Furthermore, atlases demonstrate heterogeneous performance across various tasks and an atlas derived from a fine-tuned model yields superior results for its specific application. Codes are available at https://github.com/ncclab-sustech/DCA.

## 1 Introduction

Brain atlases, as predefined parcellations that group voxel-wise fMRI signals into regions, are essential tools for reducing data dimensionality and improving interpretability in neuroimaging studies. Over the decades, hundreds of atlases have been proposed, based on anatomical [1, 2, 3], functional [4, 5, 6], and cytoarchitectonic [7] criteria. These atlases vary widely in spatial resolution (from fewer than 10 to over 1000 regions) and anatomical coverage (e.g., cortical vs. subcortical), and have become foundational resources in neuroscience.

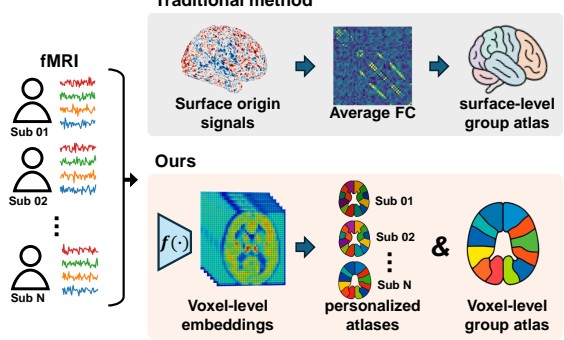

Figure 1: **Motivation**. (top) Traditional atlases cluster coarse, group-averaged functional connectivity (FC), limiting resolution and individual specificity. (bottom) DCA learns voxel-wise embeddings for personalized parcellations, enabling flexible, high-resolution group atlases.

Equal contribution: M. W., K. P., J. T.
Corresponding author: H. W., Q. L.

Despite their ubiquity, existing brain atlases suffer from several key limitations that hinder their adaptability and performance in data-driven analysis pipelines. Most atlases are built on cortical surfaces, neglecting subcortical and white-matter structures. However, growing evidence suggests that large-scale brain function emerges from interactions across the whole brain, motivating the need for voxel-based, full-brain parcellations (Fig.1). Atlas granularity is often fixed and predefined, forcing users to compromise between anatomical coverage and resolution. For example, the cortical mask in Yeo[1] spans 33k voxels per hemisphere, whereas MMP[8] covers 58k—despite nominally referring to similar regions. A more flexible framework should support arbitrary region-of-interest (ROI) selection and user-specified parcel counts. In addition, most atlases are constructed as group-level templates derived from averaged data or majority vote, which, while generalizable, overlook substantial inter-individual variability in brain function and structure. Recent studies have emphasized the value of subject-specific models in enhancing reproducibility and precision [9].

Clustering is a cornerstone of brain atlas construction, but off-the-shelf algorithms—such as K-Means [10], hierarchical clustering [11], and vanilla spectral methods [12]—are ill-suited to the characteristics of fMRI data. First, the inherently low signal-to-noise ratio of fMRI hampers these methods' ability to recover clear boundaries as the number of parcels grows. Second, even a gray-matter mask may contain tens of thousands of voxels, making the computation and storage of a full functional-connectivity matrix prohibitive. Finally, standard clustering optimizes only functional similarity and lacks any notion of spatial continuity, yielding fragmented or anatomically implausible parcels. Although one can incorporate distance-based penalties to encourage spatial contiguity, such strategies demand careful tuning lest they compromise the atlas's functional coherence.

To overcome these limitations, we introduce **Deep Cluster Atlas (DCA)**, a graph-guided deep embedding clustering framework for constructing both individualized and group-level voxel-wise brain parcellations (Fig. 1). DCA leverages a pretrained Swin-UNETR encoder to extract spatiotemporal embeddings from fMRI data and employs a spatially-regularized deep clustering module guided by a voxel-wise k-nearest-neighbor (KNN) graph. This design ensures that resulting parcels are not only functionally coherent in embedding space but also anatomically contiguous in voxel space. To systematically assess performance, we introduce a benchmarking platform (Table 8) that evaluates any input atlas using standardized internal metrics (e.g., homogeneity, silhouette coefficient) and external metrics from downstream tasks (e.g., autism diagnosis, cognitive state decoding). Across datasets and resolutions, DCA achieves 98.8% improvement in homogeneity and 29% in silhouette coefficient over existing atlases, while outperforming them in multiple classification tasks.

**Our key contributions are:**

- We propose a scalable deep clustering framework that integrates Swin-UNETR embeddings and spatial graph regularization to generate voxel-wise brain atlases.

- DCA enables flexible control over parcellation granularity and anatomical scope, supporting both personalized and group-level atlas construction.

- We release a standardized benchmarking platform to evaluate atlas quality via internal metrics and downstream tasks such as cognitive task decoding and disease diagnosis.

## 2 Related work

**MRI based brain atlas** Atlas generation based on magnetic resonance imaging (MRI) is the predominant approach for constructing parcellations. Widely adopted algorithms include k-means [10], hierarchical clustering [11], spectral clustering [12], community detection [13], normalized cuts [14], and statistical learning methods[15]. These techniques aim to maximize within-parcel homogeneity and minimize between-parcel similarity. However, functionally similar voxels are not always spatially contiguous, so enforcing spatial coherence remains challenging. To address this, spatial regularization strategies—such as Markov random field priors [2], spatially weighted clustering [14], and deep Boltzmann machine frameworks [15]—have been introduced to ensure that resulting parcels are both functionally coherent and anatomically contiguous.

**Deep clustering** Traditional clustering methods measure similarity directly in the original data space, and even when manifold-based techniques are employed, feature extraction and clustering remain two disjointed stages[16, 17, 18, 19]. Deep clustering, by contrast, integrates these steps. It

**A. Group Pretraining on Voxel-level fMRI**

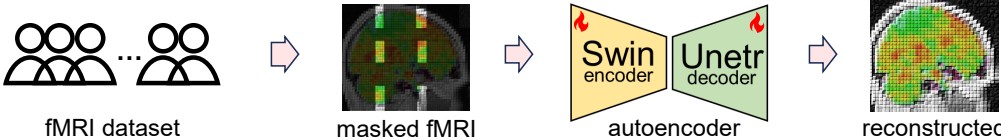

**B. Personalized Clustering with Spatial Constraints**

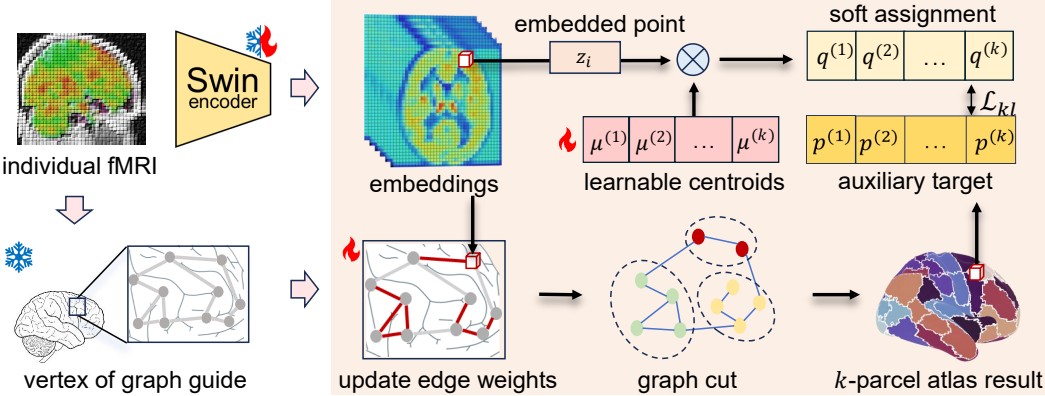

Figure 2: (A) Self-supervised pretraining of Swin-UNETR on group fMRI data: 80% of each volume is Random continuous masked in space and time and reconstructed across 1,000 resting-state trials. Fire icon denotes trainable encoder–decoder weights. (B) Personalized atlas generation at voxel resolution. Individual fMRI volumes are passed through a pretrained encoder to extract both local and global embeddings. Each voxel (red cube) is softly assigned to one of $K$ learnable centroids by measuring its distance to every centroid (top row). Simultaneously, a given ROI mask defines the nodes of a 26-neighborhood graph, whose edge weights are set to the cosine similarity between adjacent embeddings. Sparse spectral clustering on this graph produces hard auxiliary labels (bottom row). A KL-divergence loss then aligns the soft assignments with these auxiliary targets, updating both the centroids and the encoder's final projection layer. By iterating between refining embeddings, re-weighting graph edges, and updating cluster centers, the method converges to spatially contiguous, functionally coherent parcellations.

jointly refines encoder parameters and cluster centers through an auxiliary target distribution derived from the current soft assignments [20, 21]. Beyond its success in image clustering, deep clustering has proven effective for time-series segmentation [22], cell detection [23], and disease discovery [24]. When constructing a brain atlas, both functional similarity and spatial continuity must be preserved. In our framework, each voxel of a single subject serves as an individual sample for deep clustering, and we incorporate a graph-guided spatial prior to ensure contiguous parcels. To our knowledge, this is the first use of deep clustering for generating a fully continuous, voxel-level brain atlas.

**Brain segmentation**   To some extent, brain atlas construction shares conceptual similarities with semantic segmentation of brain [25, 26, 27], as both aim to partition the brain into meaningful regions. However, key differences distinguish the two tasks. First, they differ in granularity and objective: brain segmentation typically categorizes voxels based on tissue types—such as whole tumor, tumor core, and enhancing tumor—whereas brain atlases delineate functionally relevant areas, such as precentral gyrus, thalamus, hippocampus. Second, semantic segmentation is generally a supervised learning task with well-defined ground truth labels, while atlas construction is inherently unsupervised and must be evaluated using more complex criteria, such as functional or structural homogeneity. Although segmentation techniques have been employed to map existing atlases, they rely heavily on predefined templates and primarily serve to replicate rather than discover novel parcellations. For example, DDparcel assigns voxel-wise labels for 101 anatomical regions based on the Desikan–Killiany atlas, effectively reconstructing rather than redefining an atlas [28].

# 3 Methods

## 3.1 Data and preprocessing

**Atlas construction data**    We use resting-state fMRI data from 1000 subjects in the Human Connectome Project (HCP) [29] for atlas construction. All data were processed using the HCP minimal preprocessing pipeline [30], which includes gradient distortion correction, motion correction, EPI distortion correction, registration to T1-weighted images, and spatial normalization to MNI152 space. The preprocessed volumetric images were resampled to 2 mm isotropic resolution. To improve signal quality and spatial coherence, we applied spatial smoothing using AFNI's 3dBlurToFWHM [31], targeting a 3 mm FWHM. The ablation study on smoothing levels is provided in Appendix.

**Downstream task data**    For downstream evaluation, we use three public datasets: HCP, ABIDE [32], and ADNI [33]. HCP provides both resting-state and task-based fMRI data, preprocessed using the minimal preprocessing pipeline. ABIDE data come from the ABIDE I dataset, using the version preprocessed by the Preprocessed Connectomes Project (PCP) [34]. ADNI resting-state fMRI is processed using the DPABI toolbox [35], including removal of the first 10 volumes, slice timing correction, spatial normalization to the MNI152 space, smoothing with a 4 mm FWHM Gaussian kernel, linear detrending, and nuisance signal regression.

## 3.2 Personalized atlas generation

We introduce **DCA** (*Deep Clustering Atlas*), a self-supervised framework that turns a pretrained 4D encoder into a spatially consistent whole-brain fMRI parcellator (Fig. 2). Firstly, we pretrain a Swin-UNETR autoencoder on masked 4D fMRI blocks (80% spatiotemporal masking), reconstructing the missing voxels. The encoder preserves the full spatial dimensions ($H \times W \times D$), producing voxel-level embeddings that capture both local and global context (Fig. 2A). The region of interest is defined by any ROI mask. From these embeddings belong to the mask, we maintain $K$ cluster centroids. Each voxel's embedding is converted into a soft assignment by measuring its distance to every centroid (top row, Fig. 2B). In parallel, we build a 26-nearest-neighbour graph over the ROI masked voxels, weighting edges by pairwise embedding correlations. A graph-cut then yields hard auxiliary targets that enforce spatial contiguity (bottom row, Fig. 2B). Finally, we minimize the KL divergence between the soft assignments and these hard targets, backpropagating through both the encoder's final layers and the centroids. By alternating between updating embeddings and regenerating auxiliary targets, DCA converges to a voxel-level atlas that is both functionally meaningful and spatially contiguous.

**Voxel-level 4D fMRI pre-train**    To extract meaningful features from high-dimensional fMRI data, we adopt Swin-UNETR as the backbone for pretraining[26](Fig. 2A). Swin-UNETR is built upon the Swin Transformer architecture, which introduces a hierarchical structure with shifted windows to compute self-attention in a local and spatially-aware manner[36]. Unlike standard Vision Transformers (ViT) that operate on flattened global patches and often ignore local continuity[37], Swin Transformers preserve spatial hierarchies and are thus better suited for voxel-level modeling in neuroimaging. This property is particularly advantageous for fMRI, where fine-grained spatial relationships between voxels are critical.

We perform self-supervised pretraining using a masked reconstruction objective. During pretraining, we randomly mask out contiguous blocks in both space and time. Specifically, we divide the fMRI volume into non-overlapping spatiotemporal patches and then zero out 80% of those patches along the chosen spatial or temporal axis (more details in Appendix). By forcing the encoder to reconstruct missing segments using information from the surrounding unmasked regions, the model learns representations that capture both local detail and long-range dependencies across space and time. Using Swin-UNETR's encoder-decoder structure in this masked auto-encoding framework, we obtain strong voxel-wise embeddings that serve as the foundation for subsequent clustering and parcellation.

**Learnable cluster centres.**    Let $M \in \{0, 1\}^{H \times W \times D}$ be a binary mask defining the region of interest, and let $\mathcal{V} = \{i \mid M_i = 1\}$ index the $N = |\mathcal{V}|$ non-background voxels. We parameterize $K$ cluster centroids as a trainable matrix $\{\boldsymbol{\mu}_j\}_{j=1}^{K} \subset \mathbb{R}^d$, initialized with orthogonal rows and $L_2$–normalized so that $\boldsymbol{\mu}\,\boldsymbol{\mu}^{\top} = \mathbf{I}$. During each forward pass, we extract the non–background voxel

embeddings $\{\mathbf{z}_i\}_{i \in V}$ from the Swin-UNETR encoder, compute their Euclidean distances to all centroids, and convert these distances into soft assignments:

$$\Delta_{ij} = \|\mathbf{z}_i - \boldsymbol{\mu}_j\|_2, \quad w_{ij} = \exp\bigl(-\widetilde{\Delta}_{ij}\bigr), \quad \mathbf{q}_i = \frac{\mathbf{w}_i}{\sum_j w_{ij}}, \tag{1}$$

where $\widetilde{\Delta}$ denotes min–max normalization and $\mathbf{q}_i \in \Delta^{K-1}$.

**Voxel graph construction.** The vertex $\mathcal{V}$ is defined by the binary mask $M$. We extract embeddings $\{\mathbf{z}_i \in \mathbb{R}^d\}_{i \in \mathcal{V}}$ from the pretrained, fine-tunable Swin-UNETR encoder. A 26-neighbourhood graph $G = (V, E)$, which includes all voxels in a 3×3×3 cube excluding the center, is then constructed by linking each $i \in \mathcal{V}$ to its up to 26 spatial neighbours $j \in \mathcal{V}$, yielding an edge-index array $E \in \mathbb{N}^{2 \times |E|}$. Edge weights are given by the cosine similarity of demeaned embeddings:

$$a_{ij} = \cos(\mathbf{z}_i - \bar{\mathbf{z}}_i, \ \mathbf{z}_j - \bar{\mathbf{z}}_j), \tag{2}$$

where $\bar{\mathbf{z}}_i$ is the mean of $\mathbf{z}_i$. This produces a sparse adjacency $W$ with $|E| \approx 26N$ nonzero entries.

**Graph spectral clustering.** On the weighted graph $G$, we compute hard auxiliary labels $\mathbf{p} \in \{1, \ldots, K\}^{|\mathcal{V}|}$ via sparse spectral clustering. We form the unnormalized Laplacian $L = D - W$, extract the $K$ eigenvectors corresponding to the smallest eigenvalues of $L$, and finally apply K-Means on the resulting $N \times K$ embedding to assign each voxel $i$ its auxiliary label $p_i$. To preserve cluster identity across iterations, we realign each new $p_i$ to the previous labels via the Hungarian algorithm, yielding an optimal one-to-one mapping. For each iteration, given the previous labels $p_{\text{prev}}$ and the newly obtained labels $p_{\text{new}}$, we first build a cost matrix $C \in \mathbb{Z}^{K \times K}$ where

$$C_{i,j} = \sum_{n=1}^{N} \mathbb{I}\bigl(p_{\text{prev},n} = i \ \wedge \ p_{\text{new},n} = j\bigr), \tag{3}$$

i.e. the number of voxels assigned to cluster $i$ previously and cluster $j$ now. We then solve the linear assignment problem

$$\max_{\pi} \sum_{i=1}^{K} C_{i,\pi(i)}, \tag{4}$$

via the Hungarian algorithm [38] on $-C$, which yields an optimal one-to-one mapping $\pi : \{1, \ldots, K\} \to \{1, \ldots, K\}$. Finally, we relabel the new clusters $\hat{p}_i$ according to $\pi$ by $\hat{p}_i = \pi(p_i)$, thereby preserving label correspondence with earlier iterations.

**Objective.** Let $\mathbf{Q} \in (0, 1)^{N \times K}$ be the soft-assignment matrix whose $i$th row is $\mathbf{q}_i$, and let $\mathbf{P} \in \{0, 1\}^{N \times K}$ be the one-hot encoding of the aligned auxiliary labels $\hat{p}_i$. We optimize the Kullback–Leibler divergence

$$\mathcal{L} = \text{KL}\bigl(\mathbf{P} \,\|\, \mathbf{Q}\bigr) = \frac{1}{N} \sum_{i=1}^{N} \sum_{j=1}^{K} p_{ij} \log \frac{p_{ij}}{q_{ij}}, \tag{5}$$

Gradients are back-propagated only to the centroids $\{\boldsymbol{\mu}_j\}$ and the final projection block of Swin-UNETR; all other encoder weights remain fixed. This procedure jointly refines both the encoder's output and the cluster centroids to produce functionally coherent, voxel-level parcellations.

**Group atlas generation.** To facilitate downstream use and fair comparison, we also developed a streamlined procedure for deriving a group-level atlas from individual parcellations. We construct the group-level atlas in three steps as detailed in Appendix. First, we pick $K$ template label vectors, each capturing the voxel assignments of one parcel in subjects. Next, we assign every gray-matter voxel to the template vector with which it has the highest label similarity. Finally, to guarantee spatial contiguity, we keep only each parcel's largest connected component and reassign any smaller, isolated regions to adjacent parcels based on local similarity. This yields $K$ contiguous, functionally coherent parcels.

### 3.3 Atlas evaluation

To comprehensively assess the quality of any candidate atlas, we provide an interactive evaluation playground. Each input atlas is first spatially normalized to a common template space and resampled to match the reference resolution. We then perform a suite of quantitative tests, including intra-cluster similarity metrics and performance on downstream tasks. This framework enables researchers to compare and validate parcellations across multiple criteria in a standardized environment.

**Similarity metrics**  To assess the quality of brain parcellations, we adopt two evaluation metrics: global homogeneity [39, 40] and the silhouette coefficient [41]. Global homogeneity quantifies the functional coherence within each brain parcel. In our implementation, we use Pearson's correlation coefficient $R(v_i, v_j)$ between the functional time series of two voxels $v_i$ and $v_j$ as the similarity metric. The global homogeneity score $Weighted\_H$ is computed as the weighted mean within-parcel correlation across all $K$ parcels, with higher values indicating more functionally homogeneous and coherent parcellations, $m_k$ denotes the number of voxels in corresponding parcel $P_k$.

$$H = \frac{1}{m_k(m_k - 1)} \sum_{\substack{i,j \in P_k \\ i \neq j}} R(v_i, v_j), \quad Weighted\_H = \frac{\sum_{k=1}^{K}(m_k H)}{\sum_{k=1}^{K} m_k} \tag{6}$$

The silhouette coefficient measures the spatial separability and internal coherence of brain parcels. For each voxel, we compute the average dissimilarity to all other voxels within the same parcel $w_i$, and the average dissimilarity to voxels in adjacent parcels $b_i$. The dissimilarity between two voxels is defined as $1 - R$. The silhouette coefficient is then obtained by weighted averaging across all parcels, which ranges from $-1$ (poor separation) to $+1$ (excellent separation and cohesion).

$$w_i = \frac{1}{m_k - 1} \sum_{\substack{j \in P_k \\ j \neq i}} \left[1 - R(v_i, v_j)\right], \quad b_i = \frac{1}{N} \sum_{j \in \text{nb}(P_k)} \left[1 - R(v_i, v_j)\right], \quad S_i = \frac{b_i - w_i}{\max(w_i, \, b_i)}, \tag{7}$$

**Downstream tasks**  To assess the utility of brain atlases in functional modeling, we evaluate six representative downstream classification tasks spanning trait prediction, cognitive decoding, and clinical diagnosis. We use a linear support vector classifier (SVC) based on region-level functional connectivity features. Among the 12 AtlasScore benchmarks (Table 12, we selected two resting-state traits (gender [42], fluid intelligence [43]), two task-based decoding tasks [44], and two clinical diagnoses (ASD from ABIDE, AD/MCI from ADNI), while excluding tasks heavily driven by subcortical features (e.g., crystallized/general intelligence, age). This choice ensures a fair evaluation of cortex-only atlases. DCA tends to yield stronger improvements on cortex-driven tasks such as cognitive decoding, while gains on benchmarks relying more on subcortical features are less pronounced, consistent with its cortical specialization.

## 4 Implementation details

We pretrain our model using a masked reconstruction objective on fMRI data blocks of size $96 \times 96 \times 96 \times 300$, representing 3D spatial volumes with 300 temporal frames. The model is trained for 8 epochs on 2 NVIDIA A100 GPUs using a batch size of 4. The optimizer and learning rate are Adam and 0.01 respectively for both pretraining and fine-tuning. During pretraining, we adopt a masking ratio of 0.8, randomly masking 80% of the input in both spatial and temporal dimensions. The temporal length of the internal representation of Swin-UNETR is downsampled to $T = 48$, and the encoder produces a feature map of shape $96 \times 96 \times 96 \times 256$ for clustering, $d$ is 256.

For all downstream evaluations, we use data from three public datasets: HCP, ABIDE, and ADNI. All fMRI time series are masked to brain regions, detrended, and z-scored. FC matrices are computed using Pearson correlation between voxel-wise time series, and the upper triangular entries are vectorized to form feature vectors. When the feature dimension exceeds 100, we apply PCA to reduce it to 100 dimensions, balancing model complexity and sample size to ensure fair comparison. To test the efficacy of the atlas, we use a simple linear SVC for the downstream classification task.

# 5 Experimental results

To rigorously benchmark our method, we developed a comprehensive evaluation framework that assesses parcellation quality via intra- and inter-region fMRI signal correlations. Next, we apply each group atlas to a suite of real-world downstream tasks, such as cognitive task decoding to evaluate practical utility. We compare DCA against several widely used atlases: **Yeo et al.** [1] parcellates cortex into seven large-scale functional networks derived from resting-state connectivity. **Brodmann** [7] defined cortical areas based on cytoarchitectonic boundaries. **Schaefer et al.** [2] published a multi-resolution functional atlas (100–1000 parcels) using gradient-informed clustering of functional connectivity. **AAL** (Automated Anatomical Labeling) [4, 45] and **MUSE** (MUlti-atlas region Segmentation utilizing Ensembles) [46] divide the brain into hundreds of anatomy-based regions. **MMP** (Multi-Modal Parcellation) [8] integrates structural, functional, and connectivity data to define 360 cortical areas. **GIANT** (Genetically Informed brAiN aTlas) [47] achieves brain parcellation by genetically-driven integration of voxel-wise heritability and spatial proximity. Additionally, the **Watershed** Atlas and the **Allen Human Reference Atlas**-3D, 2020, were incorporated. To study the effect of granularity, we further generate DCA group atlases with $K \in \{41, 100, 200, 360, 400, 500, 800\}$ parcels constrained to the FreeSurfer gray-matter mask. The mask is derived from the cortical gray matter regions in FreeSurfer's aparc+aseg.mgz [48] and transformed into MNI152 space. In the following sections, we report correlation metrics and downstream task accuracies for each atlas configuration.

## 5.1 Main results

Fig. 3 summarizes clustering performance for several existing atlases alongside our DCA method, evaluated by Homogeneity (Fig. 3B) and Silhouette Coefficient (Fig. 3C). We sample 100 subjects from HCP randomly. Full quantitative results are given in Table 4. As the number of parcels increases from 7 to 1000, both metrics rise for all methods. To ensure a fair comparison, we match each atlas to DCA at the similar ROI numbers. Across every evaluated resolution, DCA consistently outperforms the best-performing baseline atlas, yielding higher homogeneity and silhouette scores at each scale. For example, at 200 parcels, DCA improves Homogeneity by 77.7% and Silhouette by 19.5% over the Schaefer baseline. On average, across 41 to 800 parcels, DCA improves Homogeneity by 98.8% and Silhouette by 29%, demonstrating consistent gains in functional homogeneity and spatial separation. This highlights that DCA provides a more refined and effective parcellation solution across various atlas configurations, regardless of the cluster count.

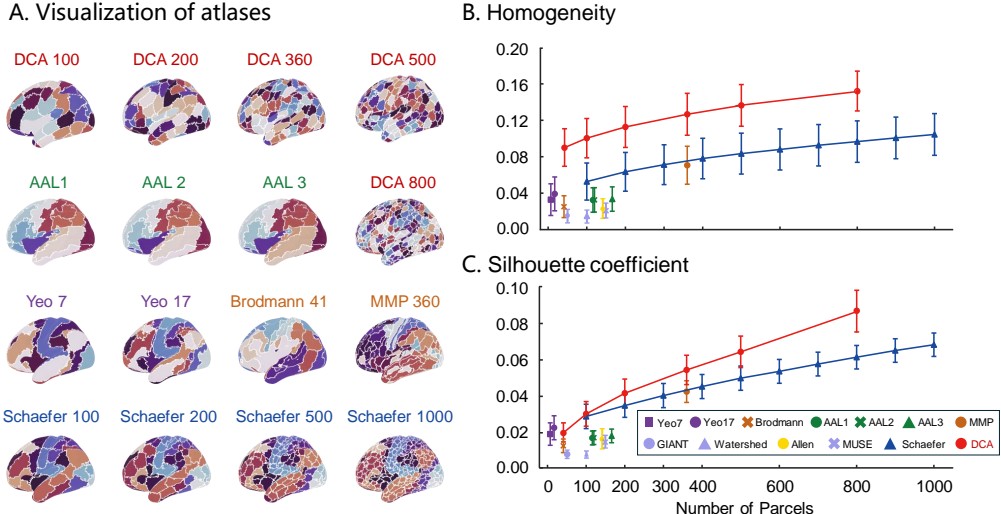

Figure 3: (A) Surface renderings of our DCA group parcellations alongside 12 mainstream atlases. (B) Homogeneity measured over 100 HCP subjects at varying numbers of parcels. (C) Silhouette coefficients for the same 100 HCP subjects and resolutions.

## 5.2 Downstream tasks

We evaluate the utility of DCA atlases on six downstream classification tasks covering behavioral prediction, cognitive decoding, and clinical diagnosis on the same 100 subjects as in the previous section (Fig. 4). Across resolutions, DCA consistently matches or outperforms the strongest baseline within each group. Full results, including extended benchmarks on additional tasks, are reported in the Appendix (Table 8).

At low resolution, DCA100 outperforms widely used atlases such as Yeo and Brodmann, which exhibit lower homogeneity and reduced classification accuracy across tasks. In medium and high resolutions, DCA200 and DCA360 achieve top performance on cognitive decoding and autism diagnosis tasks, suggesting that fine-grained voxel embeddings and spatial continuity contribute to better functional alignment than anatomically or connectivity-derived alternatives such as AAL or MMP. At ultra-high resolution (500 parcels), DCA maintains strong and stable performance across behavioral and clinical tasks. While some atlases (e.g., Schaefer500) marginally outperform DCA on individual tasks, DCA exhibits more consistent generalization across domains.

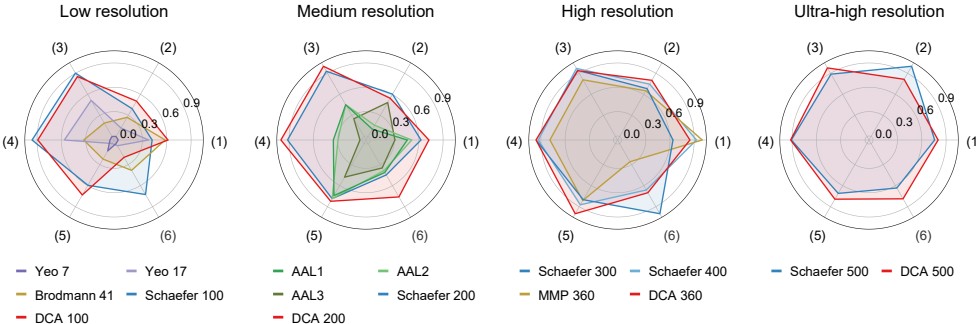

Figure 4: Performance of DCA and baseline atlases across different spatial resolutions on six downstream tasks: (1) gender prediction from resting-state FC (HCP), (2) fluid intelligence prediction from resting-state FC (HCP), (3) classification of 7 cognitive tasks from task-based FC (HCP), (4) classification of 24 cognitive tasks from task-based FC (HCP), (5) ASD vs. control classification from resting-state FC (ABIDE), (6) AD/MCI/CN classification from resting-state FC (ADNI). DCA achieves competitive or superior performance at each resolution level. Values are linearly scaled per task with 0 and 1 corresponding to the lowest and highest performing atlases, respectively.

## 5.3 Task-specific atlas

We demonstrate that our framework can be readily adapted to task-specific settings, achieving substantial improvements on the corresponding evaluation metrics. Specifically, we replace the reconstruction self-supervised Swin-UNETR with a version fine-tuned for gender classification and use its encoder to derive a task-specific atlas, denoted $DCA_{100}^{gender}$. We then aggregate fMRI signals into $K = 100$ ROIs for $N = 200$ subjects drawn from the Swin-UNETR fine-tuning test split to avoid data leakage and evaluated using two downstream classifiers:

- a compact 1-D CNN (two convolutional layers followed by two fully-connected layers),
- a graph-based $k$-GNN ($k = 2$), built from functional-connectivity graphs sparsified to the top 30 % of edges.

Both evaluations used a 70 / 10 / 20 subject split for train/validation/test, fully disjoint from fine-tuning subjects to avoid leakage. Tables S1 and S2 summarise the results. The task-adapted atlas yields consistent gains—up to **+12 %** with the CNN and **+10 %** with the $k$-GNN—while preserving spatial continuity (Table 1 and 2).

## 5.4 Ablation study

To quantify the contribution of each component, we conducted two ablation studies on 100 random subjects from HCP. First, we applied (1) K-Means clustering and (2) the same graph construction and

Table 1: CNN-based gender classification (**higher is better**).

| Atlas | Accuracy ↑ | F1 (Macro) ↑ | F1 (Weighted) ↑ |
|---|---|---|---|
| Watershed (100) | 0.73 | 0.73 | 0.73 |
| Schaefer100 | 0.65 | 0.65 | 0.65 |
| DCA100 (group) | 0.70 | 0.70 | 0.70 |
| DCA100 (individual) | 0.70 | 0.69 | 0.70 |
| $DCA_{100}^{gender}$ (group) | 0.70 | 0.67 | 0.67 |
| $DCA_{100}^{\textbf{gender}}$ **(individual)** | **0.82** | **0.82** | **0.82** |

Table 2: $k$-GNN-based gender classification (**higher is better**).

| Atlas | Accuracy ↑ | F1 (Macro) ↑ | F1 (Weighted) ↑ |
|---|---|---|---|
| Watershed (100) | 0.600 | 0.596 | 0.596 |
| Schaefer100 | 0.725 | 0.723 | 0.723 |
| DCA100 (group) | 0.650 | 0.650 | 0.650 |
| DCA100 (individual) | 0.725 | 0.716 | 0.716 |
| $DCA_{100}^{gender}$ (group) | 0.675 | 0.670 | 0.670 |
| $DCA_{100}^{\textbf{gender}}$ **(individual)** | **0.825** | **0.825** | **0.825** |

graph-cut pipeline from our method directly to the raw fMRI time series. This baseline highlights the necessity of our pretrained encoder for extracting informative features. Second, we ran both (3) K-Means and (4) the same graph cut pipeline on the encoded embeddings **without** the KL-guided joint optimization of encoder parameters and centroids, isolating the impact of our graph-guided deep clustering mechanism. In both ablations, all similarity metrics (homogeneity and silhouette) drop noticeably below those of (5) the full DCA model (Fig. 5 and Table 3). Moreover, K-Means fails to produce spatially contiguous parcels. And graph-cut improves continuity, but it still yields isolated regions. In contrast, DCA's iterative KL-driven refinement—which alternates updating graph weights, encoder parameters, and cluster centers—produces brain atlases that are highly homogeneous. We also tested the model reproducibility (Table 6 and 7), the effects of the loss (Table 15 and 20, Fig. 12) and normalization (Table 15), the smoothing (Table 16), graph cut method (Fig. 10 and Table 17), gray matter region (Table 18), number of neighbors (Table 19) and centroid initialization (Table 21) as detailed in Appendix.

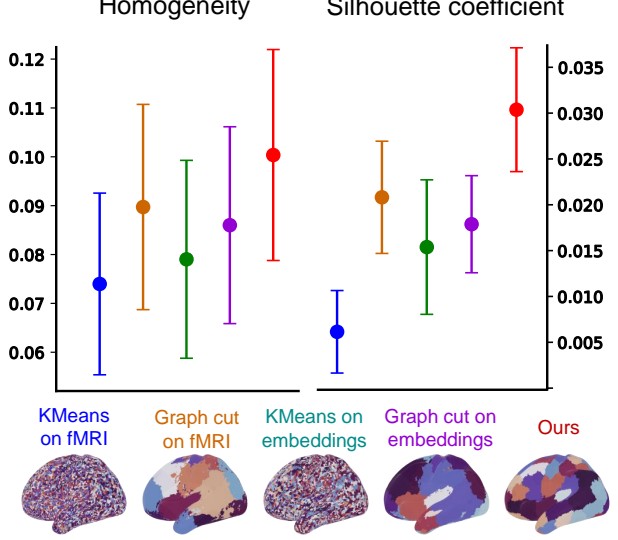

Figure 5: Directly applying K-means or graph-cut method to the raw fMRI time-series or embeddings produces parcellations with substantially lower homogeneity and silhouette scores than our iteratively optimized method. Moreover, K-means on the unprocessed signals cannot guarantee spatially contiguous regions, and while graph-cut method can partially enforce contiguity, it still fails to produce fully continuous parcels. Our method yields brain atlases that are both highly homogeneous and spatially contiguous.

Table 3: Evaluation for fMRI and embedding-based clustering with K-Means, Graph Cut, and DCA.

|  |  | Homogeneity↑ | Silhouette↑ | Connected components per parcel↓ |
|---|---|---|---|---|
| fMRI | K-Means | 0.0740±0.0186 | 0.0061±0.0045 | 447.90 |
|  | Graph Cut | 0.0860±0.0201 | 0.0179±0.0053 | 8.92 |
| Embedding | K-Means | 0.0790±0.0203 | 0.0154±0.0073 | 322.32 |
|  | Graph Cut | 0.0897±0.0210 | 0.0208±0.0061 | 4.94 |
| DCA |  | **0.1004±0.0216** | **0.0304±0.0068** | **1.0052** |

# 6 Discussion & Conclusion

In this study, we presented DCA, a unified framework for generating personalized and group-level voxel-wise brain parcellations. By combining a pretrained fMRI encoder with spatially regularized deep clustering on voxel embeddings, DCA produces anatomically contiguous and functionally meaningful atlases that capture local and global brain dynamics.

**Resolution Dependence of Downstream Tasks** We further examined how downstream task performance varies with the number of parcels. Based on these trends, tasks can be broadly grouped into three categories: (i) resolution-optimal tasks, which peak at intermediate resolutions, (ii) resolution-insensitive tasks, which remain largely stable across scales, and (iii) size-driven tasks, which monotonically track parcel granularity (Table 13). Importantly, this pattern is consistently observed for both DCA and Schaefer atlases (Table 14), indicating that there is no universally optimal resolution. Instead, the choice of parcel count should be guided by the target application. In addition, DCA offers the flexibility to further optimize atlases for specific tasks at a given resolution, as demonstrated in our task-adapted atlas experiments (Table 1 and 2). What's more, no single atlas generalizes optimally across tasks—different tasks favor different parcellations [49]—so developing task-specific atlases is warranted.

**Technical Impact** Clustering is a core operation in brain atlas construction, transforming high-dimensional neural data into interpretable regional structures. However, conventional clustering algorithms are ill-suited for the spatial and functional constraints of brain organization—they often ignore anatomical continuity, rely on coarse-grained features, and lack individual specificity. DCA addresses these challenges with a deep clustering pipeline built on a Swin-UNETR encoder pretrained for spatiotemporal representation learning. The voxel-wise embeddings are clustered using a KNN graph prior that enforces spatial smoothness, enabling anatomically contiguous regions that better reflect functional topology. A KL-divergence loss between learnable centroids and graph-induced pseudo-labels allows joint refinement of both the embedding space and clustering assignments. This design ensures that the resulting parcels are not only functionally coherent but also spatially contiguous—a critical requirement for valid brain parcellation. Our evaluation platform confirms that DCA yields superior internal consistency and downstream utility, outperforming existing atlases on homogeneity, silhouette coefficient, and classification tasks such as autism diagnosis and task decoding.

**Limitations and future directions** Despite its strengths, DCA has several limitations. First, voxel-wise representation learning and clustering incur substantial memory and computational costs, especially at whole-brain scales. Future work may explore region-specific parcellation or sparse embedding schemes to reduce overhead. Second, our reliance on fixed KNN graphs to enforce spatial continuity may inadvertently attenuate long-range functional relationships, which may suppress long-range functional relationships. Integrating adaptive or learned graphs could help balance spatial continuity with network-level functional coherence. Lastly, the current pipeline uses only single-modality fMRI data. Incorporating structural and diffusion imaging, or even electrophysiological data (e.g. sEEG), could further enhance the biological fidelity of parcellations [50, 8].Meanwhile, traditional atlas construction methods struggle to reconcile conflicting signals across modalities [51], motivating the development of a multimodal deep-clustering framework as a promising avenue for building richer, functionally and structurally grounded brain atlases.

## Acknowledgements

We thank Wenhao YE for his assistance during the rebuttal. This work was supported by the National Natural Science Foundation of China (62472206), Shenzhen Science and Technology Innovation Committee (RCYX20231211090405003, KJZD20230923115221044), Guangdong Provincial Key Laboratory of Advanced Biomaterials (2022B1212010003), and the open research fund of the Guangdong Provincial Key Laboratory of Mathematical and Neural Dynamical Systems, the Center for Computational Science and Engineering at Southern University of Science and Technology.

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

# A Appendix

## A.1 Evaluation of similarity metrics

We evaluated several atlases—including Yeo, Brodmann, GIANT, Watershed, Schaefer, AAL, Allen, MUSE, and MMP—with parcel counts ranging from 7 to 1000, using both homogeneity and the silhouette coefficient as evaluation metrics (Table 4).

Table 4: Evaluation of similarity metrics against DCA and other atlases. Values are shown as mean ± standard deviation.

| Atlas / Metrics | Yeo | | Brodmann | DCA(ours) | GIANT | Schaefer | Watershed | DCA(ours) | AAL | | Allen | MUSE | AAL |
|---|---|---|---|---|---|---|---|---|---|---|---|---|---|
| | 7 | 17 | 41 | 41 | 50 | 100 | 100 | 100 | 116 | 120 | 141 | 149 | 166 |
| Homogeneity ↑ | 0.0329 ±0.0174 | 0.0392 ±0.0186 | 0.0251 ±0.0120 | **0.0892** **±0.0204** | 0.0148 ±0.0073 | 0.0527 ±0.0204 | 0.0143 ±0.0070 | **0.1004** **± 0.0216** | 0.0324 ±0.0134 | 0.0326 ±0.0134 | 0.0230 ±0.0106 | 0.0208 ±0.0080 | 0.0335 ±0.0134 |
| Silhouette ↑ | 0.0193 ±0.0062 | 0.0228 ±0.0066 | 0.0128 ±0.0037 | **0.0198** **±0.0054** | 0.0079 ±0.0023 | 0.0290 ±0.0067 | 0.0078 ±0.0020 | **0.0304** **± 0.0068** | 0.0171 ±0.0038 | 0.0173 ±0.0038 | 0.0164 ±0.0056 | 0.0149 ±0.0035 | 0.0183 ±0.0037 |

| Schaefer | DCA(ours) | Schaefer | MMP | DCA(ours) | | Schaefer | | DCA(ours) | Schaefer | | | DCA(ours) | Schaefer | |
|---|---|---|---|---|---|---|---|---|---|---|---|---|---|---|
| 200 | 200 | 300 | 360 | 360 | 400 | 400 | 500 | 500 | 600 | 700 | 800 | 800 | 900 | 1000 |
| 0.0634 ±0.0213 | **0.1127** **± 0.0225** | 0.0712 ±0.0219 | 0.0706 ± 0.0208 | **0.1266** **± 0.0230** | **0.1294** **± 0.0230** | 0.0780 ± 0.0222 | 0.0834 ± 0.0225 | **0.1364** **± 0.0229** | 0.0880 ± 0.0226 | 0.0926 ± 0.0227 | 0.0966 ± 0.0228 | **0.1536** **± 0.0227** | 0.1006 ± 0.0229 | 0.1044 ± 0.0229 |
| 0.0349 ±0.0065 | **0.0417** **± 0.0078** | 0.0404 ±0.0066 | 0.0426 ±0.0060 | **0.0545** **± 0.0080** | **0.0572** **± 0.0082** | 0.0454 ±0.0066 | 0.0500 ±0.0067 | **0.0644** **± 0.0086** | 0.0537 ±0.0065 | 0.0577 ±0.0065 | 0.0615 ±0.0065 | **0.0866** **± 0.0114** | 0.0652 ±0.0065 | 0.0683 ± 0.0064 |

## A.2 Cross-dataset generalization on CHCP

We further assess our model's cross-dataset generalization by applying the Swin-UNETR encoder—pretrained on the HCP dataset—to individual atlas generation on the other independent dataset (Fig. 6 and Table 5). The Chinese Human Connectome Project (CHCP) dataset comprises high-resolution multimodal MRI—including structural, diffusion, and resting-state fMRI—from healthy Chinese adults, and uses the same acquisition parameters and HCP preprocessing pipeline [52]. Our results demonstrate that, without any additional fine-tuning, DCA produces coherent, spatially contiguous parcellations on CHCP dataset that surpass other atlases in both metrics, underscoring the robustness of the learned voxel embeddings.

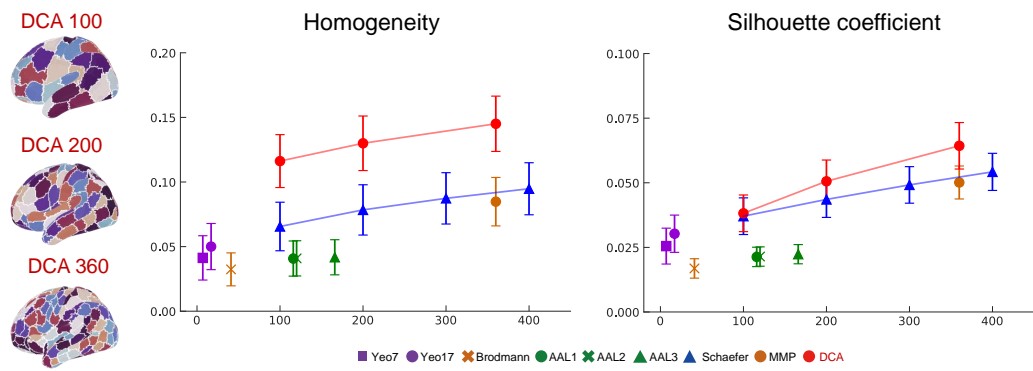

Figure 6: Homogeneity and Silhouette coefficients measured over 100 CHCP subjects at varying numbers of parcels.

## A.3 Atlas of the subcortex and white matter

Because our method learns voxel-level embeddings across the entire brain, it is applicable to any arbitrary brain structure. Given a region-of-interest (ROI) mask, our framework can generate parcellations at a specified resolution. Here, we demonstrate two applications: atlas construction for the subcortex and white matter (Fig. 7). The corresponding ROI masks are extracted from FreeSurfer's aparc+aseg.mgz [48] and transformed into MNI152 space. We evaluate parcellations with $\{10, 20, 40, 50\}$ clusters for the subcortex and $\{50, 100, 200, 400\}$ clusters for the white matter.

Table 5: Evaluation of similarity metrics against DCA and other atlases on the CHCP dataset

| Metrics \ Atlas | Yeo | | Brodmann | Schaefer | **DCA** | AAL | |
|---|---|---|---|---|---|---|---|
| | 7 | 17 | 41 | 100 | 100 | 116 | 120 |
| Homogeneity ↑ | 0.0413 ±0.0171 | 0.0500 ±0.0179 | 0.0324 ±0.0127 | 0.6560 ±0.0188 | **0.1162 ± 0.0205** | 0.0407 ±0.0136 | 0.0409 ±0.0136 |
| Silhouette ↑ | 0.0255 ±0.0069 | 0.0303 ±0.0072 | 0.0169 ±0.0038 | 0.0371 ±0.0071 | **0.0382 ± 0.0071** | 0.0213 ±0.0038 | 0.0215 ±0.0037 |

| Metrics \ Atlas | AAL | Schaefer | **DCA** | Schaefer | MMP | **DCA** | Schaefer |
|---|---|---|---|---|---|---|---|
| | 166 | 200 | 200 | 300 | 360 | 360 | 400 |
| Homogeneity ↑ | 0.0418 ±0.0136 | 0.0784 ±0.0195 | **0.1300 ± 0.0211** | 0.0873 ±0.0199 | 0.0848 ±0.0187 | **0.1451 ±0.0214** | 0.0948 ±0.0201 |
| Silhouette ↑ | 0.0224 ±0.0037 | 0.0435 ±0.0069 | **0.0506 ± 0.0082** | 0.0492 ±0.0071 | 0.0501 ±0.0064 | **0.0643 ±0.009** | 0.0542 ±0.0072 |

In future work, incorporating additional structural information, such as white matter fiber orientations, could further improve the quality of the parcellations.

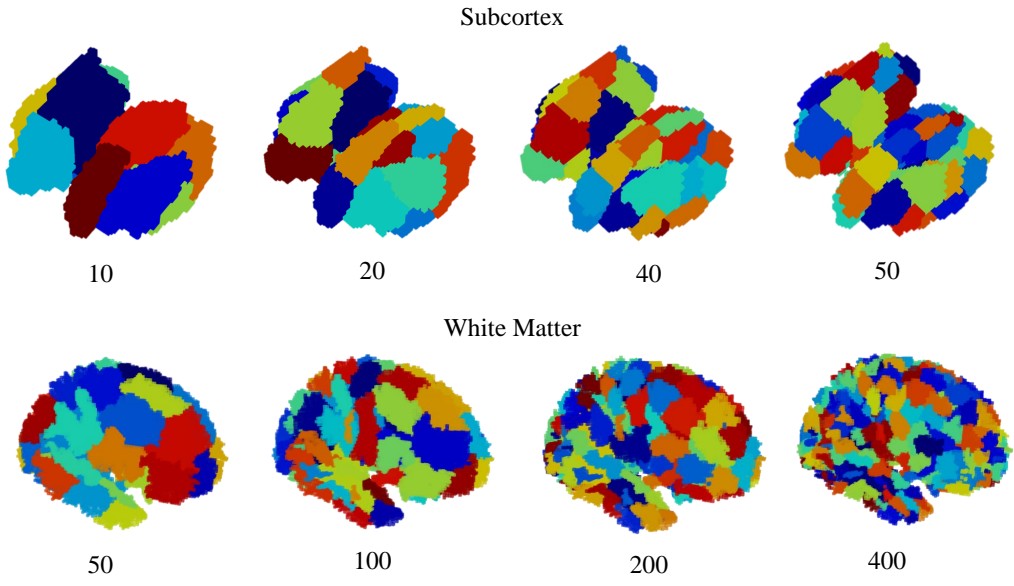

Subcortex

10    20    40    50

White Matter

50    100    200    400

Figure 7: Voxel-wise parcellations of subcortical and white matter regions under varying granularity levels.

## A.4 Model reproducibility

To quantify variability across repeated executions, we ran the complete pipeline five times on the same fMRI segment and compared the resulting parcellations using Dice, intersection-over-union (IoU), voxel assignment consistency (VAC; the fraction of voxels that keep the same label after Hungarian alignment), adjusted Rand index (ARI), and normalized mutual information (NMI) (Table 6). Variability stems chiefly from (i) the random initialization of cluster centroids in the spectral-graph step and (ii) the initialization of the model's centroid matrix. With a fixed random seed, the atlas is perfectly deterministic. Because the loss contains a Kullback–Leibler (KL) term that aligns model assignments to graph-clustering assignments, reproducible results require fixing

both seeds or fixing one seed and deriving the other from it (e.g., initializing the graph centroids with the model's centroid matrix). Under realistic stochasticity from spectral clustering and model initialization, more than $80\%$ of voxels retain their labels across runs; disagreements are typically confined to regions where a dominant parcel can be subdivided into finer clusters.

Beyond seed control, stability can be boosted by seeding cluster centroids with an external prior—such as a population template—then allowing DCA to refine these priors into subject-specific atlases. This strategy preserves the anatomical grounding of established atlases while exploiting DCA's capacity for individualized refinement.

Table 6: Model reproducibility across five runs.

|  | Dice ↑ | IoU ↑ | VAC ↑ | ARI ↑ | NMI ↑ |
|---|---|---|---|---|---|
| Both–Fixed | $1.000 \pm 0.000$ | $1.000 \pm 0.000$ | $1.000 \pm 0.000$ | $1.000 \pm 0.000$ | $1.000 \pm 0.000$ |
| Model seed–Fixed | $0.809 \pm 0.025$ | $0.737 \pm 0.031$ | $0.809 \pm 0.025$ | $0.748 \pm 0.030$ | $0.904 \pm 0.009$ |
| Graph seed–Fixed | $0.832 \pm 0.020$ | $0.769 \pm 0.027$ | $0.845 \pm 0.022$ | $0.808 \pm 0.027$ | $0.921 \pm 0.009$ |
| Both–Random | $0.822 \pm 0.023$ | $0.753 \pm 0.030$ | $0.823 \pm 0.021$ | $0.768 \pm 0.025$ | $0.911 \pm 0.009$ |
| Null | $0.500 \pm 0.017$ | $0.352 \pm 0.016$ | $0.500 \pm 0.017$ | $0.386 \pm 0.011$ | $0.747 \pm 0.005$ |

While the method produces consistent atlases when run multiple times on the same fMRI segment with identical settings, we then evaluated cross-segment stability. To quantify consistency, we produced $\mathrm{DCA}_{100}$ atlases from ten non-overlapping fMRI segments for each of ten HCP participants, spanning multiple runs and both phase-encoding directions. Atlas similarity was measured with Dice and intersection-over-union (IoU). As summarised in Table 7, intra-subject similarity is markedly higher than both inter-subject similarity and a null baseline obtained by randomly partitioning the cortical mask into 100 spatially contiguous, equal-sized parcels (with parcel correspondence solved via the Hungarian algorithm). These results confirm that DCA yields reproducible atlases despite stochastic initialisation.

Table 7: Intra- and inter-subject atlas similarity.

|  | **Dice ↑** | **IoU ↑** |
|---|---|---|
| Inter-subject (null) | 0.497 | 0.349 |
| Inter-subject | **0.614** | **0.481** |
| Intra-subject (null) | 0.506 | 0.358 |
| Intra-subject | **0.789** | **0.707** |

## A.5  Group atlas generation

To facilitate downstream use and fair comparison, we developed a streamlined procedure to construct a group-level atlas from individual parcellations (Algorithm 1). Given $N$ subject-specific atlases with a common label system of $K$ parcels, we generate a spatially contiguous group-level atlas through three steps:

**Step 1: Template label selection.**  We first identify a set of reliable voxels $\mathcal{V}_{\mathrm{core}}$ that are absent in at most $\alpha \cdot N$ subjects (we use $\alpha = 0.2$). For each voxel $v \in \mathcal{V}_{\mathrm{core}}$, we form a label vector $\mathbf{z}_v = [A^{(1)}(v), A^{(2)}(v), \ldots, A^{(N)}(v)]$, where $A^{(i)}(v)$ denotes the label assigned to voxel $v$ in subject $i$. We sort all such vectors by frequency of occurrence, and select the top $K$ vectors that have pairwise Hamming distance less than $\beta \cdot N$ (we use $\beta = 0.2$) as the *template label vectors* $\{\mathbf{t}_1, \ldots, \mathbf{t}_K\}$.

**Step 2: Voxel-to-template assignment.**  We then identify $\mathcal{V}_{\mathrm{assign}}$, the set of voxels absent in at most $\gamma \cdot N$ subjects (with $\gamma = 0.8$), and assign each $v \in \mathcal{V}_{\mathrm{assign}}$ to the template $k$ that maximizes the agreement:

$$L(v) = \arg\max_k \sum_{i=1}^{N} \mathbb{I}[A^{(i)}(v) = t_k^{(i)}]$$

This results in a $K$-label volumetric map $\mathbf{L}$ that is not necessarily spatially contiguous.

**Step 3: Spatial contiguity enforcement.** For each parcel $k$, we retain only its largest 6-connected component, and mark all other voxels in $k$ as unlabeled. We then iteratively reassign these dropped voxels as follows: for each unlabeled voxel $v$ with at least one labeled 6-connected neighbor, we compute the Hamming distance between $\mathbf{z}_v$ and the label vectors $\{\mathbf{z}_u\}$ of its labeled neighbors $u \in \mathcal{N}_6(v)$. The voxel $v$ is then assigned the same label as the neighbor $u^*$ with the smallest distance, i.e., $L(v) = L(u^*)$. This process repeats until all voxels are labeled, resulting in a group-level atlas with $K$ spatially contiguous and functionally consistent regions.

---

**Algorithm 1** Group-level atlas generation from individual parcellations

---

**Require:** Subject-wise atlases $\{\mathbf{A}^{(1)}, \mathbf{A}^{(2)}, \dots, \mathbf{A}^{(N)}\}$, number of parcels $K$, thresholds $\alpha, \beta, \gamma$
**Ensure:** Group-level atlas $\mathbf{L}$
 1: Identify $\mathcal{V}_{\text{core}} \leftarrow \{v : \text{voxel absent in } \leq \alpha N \text{ subjects}\}$
 2: For each $v \in \mathcal{V}_{\text{core}}$, construct label vector $\mathbf{z}_v = [A^{(1)}(v), \dots, A^{(N)}(v)]$
 3: Count frequency of each $\mathbf{z}_v$; sort descending
 4: Select top $K$ vectors $\{\mathbf{t}_1, \dots, \mathbf{t}_K\}$ with pairwise Hamming distance $\leq \beta N$
 5: Identify $\mathcal{V}_{\text{assign}} \leftarrow \{v : \text{voxel absent in } \leq \gamma N \text{ subjects}\}$
 6: **for all** $v \in \mathcal{V}_{\text{assign}}$ **do**
 7:     Assign $L(v) \leftarrow \arg\max_k \sum_{i=1}^{N} \mathbb{I}[A^{(i)}(v) = t_k^{(i)}]$
 8: **end for**
 9: **for all** label $k = 1$ to $K$ **do**
10:     Keep the largest 6-connected component of label $k$
11:     Mark all other voxels in $k$ as `unlabeled`
12: **end for**
13: **while** any voxel is `unlabeled` **do**
14:     **for all** unlabeled voxel $v$ **do**
15:         **if** $v$ has at least one labeled 6-neighbor **then**
16:             Find $u^* = \arg\min_{u \in \mathcal{N}_6(v), \text{ labeled}} \text{Hamming}(\mathbf{z}_v, \mathbf{z}_u)$
17:             Assign $L(v) \leftarrow L(u^*)$
18:         **end if**
19:     **end for**
20: **end while**

---

## A.6 AtlaScore: atlas evaluation platform

Due to space constraints, we have omitted the full set of results from the main text. In the Supplementary Material, we present detailed experimental designs and complete outcomes for 3 similarity evaluations and 12 downstream tasks (Table 12).

In addition to the 12 downstream tasks provided by AtlaScore, we further evaluated atlas utility using modern neural classifiers. Specifically, we trained a compact 1-D CNN on ROI-level time series (Table 1) and a graph-based $k$-GNN on FC graphs (Table 2). These models provide a complementary perspective by directly testing whether atlas parcellations facilitate feature extraction for nonlinear learning. These findings reinforce that DCA not only benefits classical SVC pipelines but also enhances performance in deep learning–based settings.

### A.6.1 Distance-controlled boundary coefficient (DCBC)

In the main text, we have introduced Homogeneity and Silhouette in Section 3. Such conventional metrics overlook the intrinsic spatial smoothness of brain signals, often overestimating parcellation quality by conflating spatial proximity with functional similarity, especially in high-resolution cortical data, where false boundaries may emerge due to smoothness rather than genuine functional distinctions [50]. To mitigate this bias, DCBC groups vertex pairs based on their spatial separation and evaluates functional similarity differences between within- and between-parcel pairs at each distance level. The DCBC metric is formally defined as follows:

Table 8: Experiment index, name, and description.

| # | Name | Description |
|---|------|-------------|
| 1 | Similarity–Homogeneity | Measure mean intra-cluster correlation |
| 2 | Similarity–Silhouette | Compute silhouette coefficient over voxels |
| 3 | Similarity–DCBC A.6.1 | Evaluating brain parcellations using the distance-controlled boundary coefficient [50] |
| 4 | Downstream–Gender classification A.6.2 | Predict biological sex using FC |
| 5 | Downstream–Fluid Intelligence A.6.3 | Predict fluid intelligence level using FC |
| 6 | Downstream-Cognitive task (7-way) A.6.4 | Classify 7 cognitive tasks using FC |
| 7 | Downstream-Cognitive task (24-way) A.6.4 | Classify 24 cognitive tasks using FC |
| 8 | Downstream-Autism diagnosis A.6.5 | Classify autism vs. healthy controls using FC |
| 9 | Downstream-AD diagnosis A.6.6 | Classify AD / MCI / CN using FC |
| 10 | Downstream-FC stability A.6.7 | Within-subject FC similarity |
| 11 | Downstream-Fingerprinting A.6.8 | Subject identification via FC matching |
| 12 | Downstream-Age group classification A.6.9 | Predict age group labels |
| 13 | Downstream-Crystallized intelligence A.6.3 | Predict crystallized intelligence level using FC |
| 14 | Downstream-General intelligence A.6.3 | Predict overall cognitive ability level using FC |
| 15 | Downstream-Autism cross-site A.6.5 | Cross-site classification of autism vs. healthy controls using FC |
| 16 | Downstream-Gender classification (CNN) A.6.10 | Predict biological sex using time series |
| 17 | Downstream-Gender classification ($k$-GNN) A.6.11 | Predict biological sex using FC |

$$\text{DCBC} = \sum_{i=1}^{N} w_i d_i, \tag{8}$$

where per-bin correlation difference $d_i = \text{corr}_{\text{within}}(i)$ and $\text{corr}_{\text{between}}(i)$ are the mean functional correlations of within-parcel and between-parcel vertex pairs in the i-th spatial distance bin, respectively; The variance $var(d_i)$ reflects how vertex pair counts $(n_{w,i}, n_{b,i})$ affect the reliability of $d_i$ in each spatial bin, while the precision weights $w_i$ subsequently compensate for this uncertainty by assigning higher influence to bins with lower variance during DCBC computation. $var(d_i)$ and $w_i$ take the following forms:

$$var(d_i) = \frac{n_{w,i} + n_{b,i}}{n_{w,i} n_{b,i}}, w_i = \frac{\dfrac{n_{w,i}\, n_{b,i}}{n_{w,i} + n_{b,i}}}{\displaystyle\sum_{j=1}^{N} \frac{n_{w,j}\, n_{b,j}}{n_{w,j} + n_{b,j}}}. \tag{9}$$

By controlling for spatial distance in this way, DCBC disentangles true functional boundaries from artifacts of spatial smoothness, providing a more reliable parcellation assessment. In our evaluation process, all parameters followed the settings specified in [50]. However, we note that DCBC was mainly developed for surface-based parcellations and becomes computationally prohibitive at the fine voxel resolution employed by DCA. Therefore, we computed DCBC scores by projecting volumetric atlases onto the cortical surface (fsLR 32k template) [53], analyzing only data from the left hemisphere. This surface-based approach exceeds the scope of our native volumetric framework.

### A.6.2 Gender classification

We evaluated the ability of each atlas to support gender classification based on resting-state functional connectivity (FC). We used data from 100 unrelated subjects in the Human Connectome Project (HCP) [29], each contributing multiple FC samples constructed from 300 consecutive TRs of resting-state

fMRI. For each atlas, FC matrices were computed and their upper-triangular entries were used as features.

To ensure subject-level generalization, we performed 10-fold cross-validation across subjects: in each fold, 90 subjects were used for training and 10 for testing. A linear support vector classifier (SVC) was trained on the training set. If the FC dimensionality exceeded 100, we applied principal component analysis (PCA) to reduce dimensionality: features were projected onto the top 100 principal components computed from the training data, and test samples were projected into the same subspace. Classification accuracy on the test subjects was recorded for each fold and averaged to obtain the final performance.

### A.6.3 Fluid, crystallized, and general intelligence level prediction

We evaluated whether atlas-based FC features can predict individual differences in fluid, crystallized, and general intelligence. We used the corresponding HCP behavioral score (CogFluidComp_AgeAdj, CogCrystalComp_AgeAdj, and CogTotalComp_AgeAdj) to define three classes: low (<85), medium (85–115), and high (>115) intelligence. Each subject contributed multiple FC samples from resting-state fMRI (300 TRs per sample). 10-fold cross-validation was performed across subjects, using a linear SVC. When the number of FC features exceeded 100, PCA was applied to project the data onto the top 100 principal components computed from the training set.

### A.6.4 Cognitive task classification

We assessed whether atlas-based FC can distinguish different cognitive states using task-fMRI data from 100 HCP subjects. For the 7-class classification, each subject completed seven tasks—working memory, gambling, motor, language, social, relational, and emotional—each contributing one FC matrix. For the 24-class classification, we further segmented each task into its constituent conditions (e.g., 0-back faces, math, fear), resulting in 24 fine-grained task labels. Each subject contributed one FC matrix per task condition.

We trained a linear SVC to predict the task label from FC features using 10-fold cross-validation across subjects. When the number of FC features exceeded 100, PCA was applied to project the data onto the top 100 principal components computed from the training set.

### A.6.5 Autism diagnosis and cross-site generalization

We evaluated whether FC features can distinguish individuals with autism spectrum disorder (ASD) from healthy controls using resting-state fMRI data from the ABIDE dataset (n = 871) [32]. For each subject, an FC matrix was computed and used to train a linear SVC for binary classification. We considered two evaluation settings. In the first, we performed 10-fold cross-validation at the subject level to assess within-dataset classification performance. In the second, we tested cross-site generalization by holding out one acquisition site for testing while training on subjects from all other sites, repeating this procedure across all sites. When the number of FC features exceeded 100, PCA was applied to project the data onto the top 100 principal components computed from the training set.

### A.6.6 AD diagnosis

We evaluated the ability of FC features to classify individuals into Alzheimer's disease (AD), mild cognitive impairment (MCI), or cognitively normal (CN) groups. We used resting-state fMRI data from 267 subjects in the ADNI dataset [33], each labeled as AD, MCI, or CN. FC matrices were computed for each subject and used to train a linear SVC for 3-class classification. 10-fold cross-validation was performed across subjects. When the number of FC features exceeded 100, PCA was applied to project the data onto the top 100 principal components computed from the training set.

### A.6.7 FC stability

We evaluated the within-subject stability of FC to assess how consistently an atlas captures individual functional architecture [54]. We used resting-state fMRI data from 100 HCP subjects. Each fMRI scan was segmented into multiple non-overlapping windows of 300 TRs, and an FC matrix was computed per window. For each subject, we calculated the Pearson correlation between the vectorized upper triangles of all FC matrix pairs and averaged the resulting values to obtain a single stability

score. These scores were then aggregated across all subjects to evaluate the group-level FC stability supported by each atlas.

### A.6.8 Fingerprinting

To evaluate how well an atlas captures individual-specific features in FC, we conducted a subject identification (fingerprinting) task [55]. We used resting-state fMRI data from 100 subjects, each providing multiple FC matrices. For each subject, one FC was randomly selected as the reference. The remaining FCs were matched to all reference FCs by computing the Pearson correlation between the vectorized upper triangle of each pair. A prediction was considered correct if the most highly correlated reference FC belonged to the same subject as the query FC. Each subject's identification accuracy was computed, and group-level performance was obtained by averaging across subjects.

### A.6.9 Age group classification

We evaluated whether atlas-based FC features can predict individual differences in age group. We used the HCP-provided Age variable to assign subjects into three age groups: 21–25, 26–30, and 31 years or older. Each subject contributed multiple FC samples from resting-state fMRI (300 TRs per sample). 10-fold cross-validation was performed across subjects, using a linear SVC. When the number of FC features exceeded 100, PCA was applied to project the data onto the top 100 principal components computed from the training set.

### A.6.10 Gender classification (CNN)

We further evaluated atlas performance on a CNN-based gender classification task. We used ROI-level resting-state fMRI time series from the HCP dataset. The classifier was a compact 1-D convolutional neural network, consisting of two convolutional layers followed by two fully connected layers, trained end-to-end to predict gender from the input time series. To avoid subject leakage, data were split at the subject level into separate train, validation, and test sets, and all atlases were evaluated under the same protocol.

### A.6.11 Gender classification ($k$-GNN)

We also evaluated atlas performance on a graph-based gender classification task using a $k$-GNN classifier with $k = 2$. Following the protocol in [56], we constructed a base graph where each node corresponds to an ROI, and weighted edges were defined by functional connectivity (FC). The graph was sparsified by retaining the top 30% of edges by magnitude, and the resulting graph was fed to a standard 2-GNN with a readout operation to obtain subject-level embeddings, which were then passed to a linear classifier. Training used a 70/10/20 subject-level split for train/validation/test, ensuring no subject leakage. All atlases were evaluated under the same protocol.

## A.7 Complete DCA performance on AtlasScore

### A.7.1 Evaluation on different smoothing levels

In the main text, we evaluated similarity-related metrics using data smoothed with a 3mm FWHM kernel. Here, we additionally report the results obtained with unsmoothed data and data smoothed with a 6mm FWHM kernel. The overall conclusions remain consistent across different smoothing levels (Fig. 8, Table 9, and Table 10).

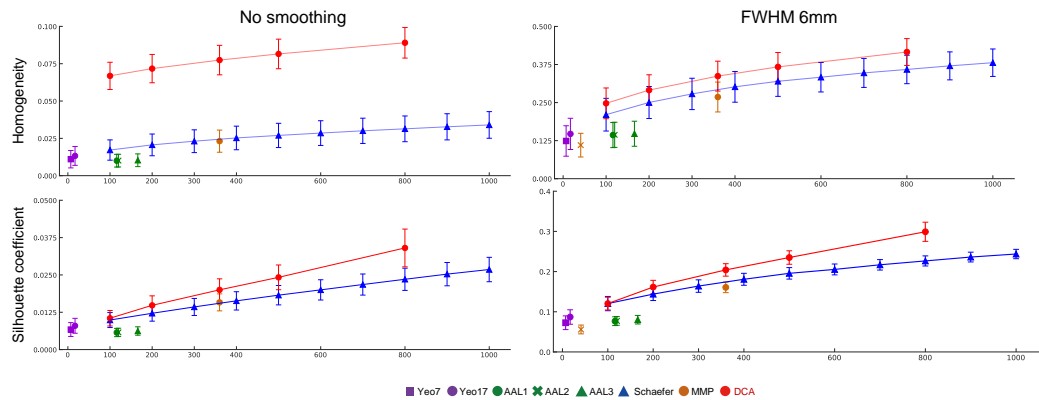

Figure 8: Homogeneity and silhouette were measured over 100 HCP subjects on different smoothing levels.

Table 9: Evaluation on no smoothing data.

| Metrics \ Atlas | Yeo | | Brodmann | Schaefer | DCA | AAL | | | Schaefer | DCA | Schaefer |
|---|---|---|---|---|---|---|---|---|---|---|---|
| | 7 | 17 | 41 | 100 | 100 | 116 | 120 | 166 | 200 | 200 | 300 |
| Homogeneity ↑ | 0.0110 ±0.0058 | 0.0133 ±0.0063 | - | 0.0172 ±0.0068 | **0.0669** ±**0.0091** | 0.0100 ±0.0043 | 0.0101 ±0.0043 | 0.0103 ±0.0043 | 0.0206 ±0.0073 | **0.0717** ±**0.0095** | 0.0231 ±0.0076 |
| Silhouette ↑ | 0.0067 ±0.0023 | 0.0080 ±0.0025 | - | 0.0099 ±0.0025 | **0.0105** ±**0.0026** | 0.0057 ±0.0014 | 0.0058 ±0.0014 | 0.0062 ±0.0014 | 0.0122 ±0.0026 | **0.0148** ±**0.0032** | 0.0143 ±0.0029 |

| Metrics \ Atlas | MMP | DCA | Schaefer | | DCA | Schaefer | | | DCA | Schaefer | |
|---|---|---|---|---|---|---|---|---|---|---|---|
| | 360 | 360 | 400 | 500 | 500 | 600 | 700 | 800 | 800 | 900 | 1000 |
| Homogeneity ↑ | 0.0231 ±0.0074 | **0.0774** ±**0.0099** | 0.0253 ±0.0079 | 0.0270 ±0.0081 | **0.0815** ±**0.0099** | 0.0285 ±0.0083 | 0.0301 ±0.0085 | 0.0314 ±0.0086 | **0.0890** ±**0.0103** | 0.0327 ±0.0088 | 0.0340 ±0.0089 |
| Silhouette ↑ | 0.0158 ±0.0028 | **0.0200** ±**0.0037** | 0.0163 ±0.0031 | 0.0182 ±0.0032 | **0.0242** ±**0.0041** | 0.0200 ±0.0034 | 0.0218 ±0.0035 | 0.0235 ±0.0037 | **0.0340** ±**0.0063** | 0.0253 ±0.0039 | 0.0268 ±0.0041 |

Table 10: Evaluation on 6mm FWHM smoothed data.

| Metrics \ Atlas | Yeo | | Brodmann | Schaefer | DCA | AAL | | | Schaefer | DCA | Schaefer |
|---|---|---|---|---|---|---|---|---|---|---|---|
| | 7 | 17 | 41 | 100 | 100 | 116 | 120 | 166 | 200 | 200 | 300 |
| Homogeneity ↑ | 0.1239 ±0.0500 | 0.1472 ±0.0512 | 0.1102 ±0.0389 | 0.2102 ±0.0537 | **0.2475** ±**0.0504** | 0.1434 ±0.0413 | 0.1441 ±0.0413 | 0.1476 ±0.0411 | 0.2502 ±0.0526 | **0.2909** ±**0.0503** | 0.2784 ±0.0515 |
| Silhouette ↑ | 0.0727 ±0.0171 | 0.0870 ±0.0180 | 0.0560 ±0.0109 | 0.1204 ±0.0177 | **0.1203** ±**0.0159** | 0.0766 ±0.0109 | 0.0773 ±0.0108 | 0.0803 ±0.0106 | 0.1439 ±0.0159 | **0.1615** ±**0.0168** | 0.1639 ±0.0155 |

| Metrics \ Atlas | MMP | DCA | Schaefer | | DCA | Schaefer | | | DCA | Schaefer | |
|---|---|---|---|---|---|---|---|---|---|---|---|
| | 360 | 360 | 400 | 500 | 500 | 600 | 700 | 800 | 800 | 900 | 1000 |
| Homogeneity ↑ | 0.2682 ±0.0491 | **0.3368** ±**0.0490** | 0.3016 ±0.0504 | 0.3198 ±0.0494 | **0.3671** ±**0.0472** | 0.3332 ±0.0484 | 0.3473 ±0.0476 | 0.3587 ±0.0467 | **0.4159** ±**0.0441** | 0.3705 ±0.0460 | 0.3809 ±0.0451 |
| Silhouette ↑ | 0.1608 ±0.0132 | **0.2041** ±**0.0155** | 0.1808 ±0.0149 | 0.1956 ±0.0143 | **0.2350** ±**0.0168** | 0.2054 ±0.0135 | 0.2169 ±0.0130 | 0.2265 ±0.0126 | **0.2990** ±**0.0238** | 0.2361 ±0.0122 | 0.2437 ±0.0117 |

### A.7.2   DCBC performance across Atlases

DCBC was mainly developed for surface-based parcellations and becomes computationally prohibitive at the fine voxel resolution employed by DCA. Therefore, we computed DCBC scores by projecting volumetric atlases onto the cortical surface (fsLR 32k template [53]), analyzing only data from the left hemisphere. This surface-based approach exceeds the scope of our native volumetric framework (Fig. 9 and Table 11 ).

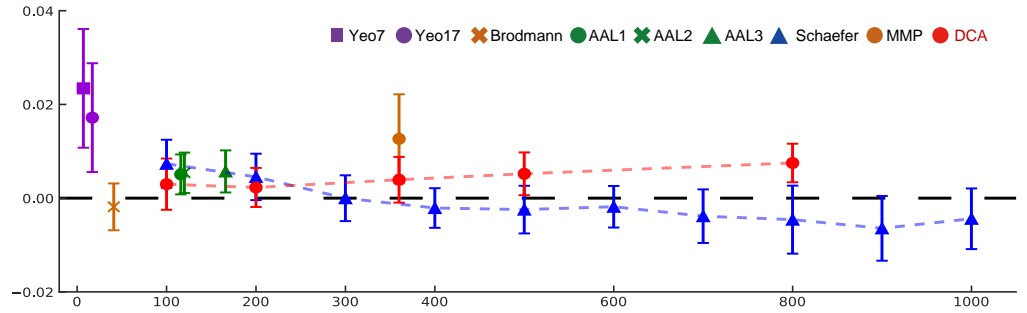

Figure 9: DCBC measured over 100 HCP subjects at varying numbers of parcels.

Table 11: DCBC values and standard deviations for various atlases.

| Atlas | #Parcels | mean | std. |
|---|---:|---:|---:|
| Yeo7 | 7 | 0.0234 | 0.0127 |
| Yeo17 | 17 | 0.0172 | 0.0116 |
| Brodmann | 41 | -0.0019 | 0.0050 |
| Schaefer | 100 | 0.0073 | 0.0051 |
| DCA | 100 | 0.0030 | 0.0055 |
| AAL 1 | 116 | 0.0051 | 0.0043 |
| AAL 2 | 120 | 0.0054 | 0.0043 |
| AAL 3 | 166 | 0.0057 | 0.0045 |
| Schaefer | 200 | 0.0045 | 0.0050 |
| DCA | 200 | 0.0023 | 0.0042 |
| Schaefer | 300 | -0.0000 | 0.0049 |
| MMP | 360 | 0.0127 | 0.0095 |
| DCA | 360 | 0.0039 | 0.0049 |
| Schaefer | 400 | -0.0021 | 0.0042 |
| Schaefer | 500 | -0.0024 | 0.0051 |
| DCA | 500 | 0.0052 | 0.0046 |
| Schaefer | 600 | -0.0018 | 0.0044 |
| Schaefer | 700 | -0.0038 | 0.0057 |
| Schaefer | 800 | -0.0046 | 0.0073 |
| DCA | 800 | 0.0075 | 0.0041 |
| Schaefer | 900 | -0.0064 | 0.0069 |
| Schaefer | 1000 | -0.0044 | 0.0065 |

### A.7.3 Downstream task performance across atlases

Due to space constraints, only a subset of results was included in the main text. Here, we provide the complete evaluation on all 12 downstream tasks across 16 atlases (Table 12). Results are reported as mean ± standard deviation, averaged over 10-fold cross-validation or subject-level evaluation. Within each resolution group, the best-performing atlas for each task is highlighted in bold.

Table 12: Evaluation of downstream task performance across atlases. Values are shown as mean ± standard deviation.

| Task | Yeo 7 | Yeo 17 | Brodmann 41 | Schaefer 100 | DCA 100 | AAL 116 | AAL 120 | AAL 166 | Schaefer 200 | DCA 200 | Schaefer 300 | MMP 360 | DCA 360 | Schaefer 400 | Schaefer 500 | DCA 500 |
|---|---|---|---|---|---|---|---|---|---|---|---|---|---|---|---|---|
| Gender classification | 0.547 ±0.077 | 0.620 ±0.063 | 0.659 ±0.078 | 0.628 ±0.053 | **0.666 ±0.080** | 0.636 ±0.052 | 0.646 ±0.055 | 0.606 ±0.102 | 0.668 ±0.042 | **0.687 ±0.073** | 0.670 ±0.067 | **0.740 ±0.065** | 0.710 ±0.059 | 0.726 ±0.059 | 0.694 ±0.066 | **0.702 ±0.079** |
| Fluid intelligence | 0.415 ±0.032 | 0.433 ±0.072 | 0.456 ±0.046 | 0.474 ±0.080 | **0.491 ±0.082** | 0.431 ±0.052 | 0.439 ±0.065 | 0.488 ±0.081 | **0.505 ±0.078** | 0.497 ±0.074 | 0.517 ±0.082 | 0.513 ±0.087 | **0.535 ±0.084** | 0.527 ±0.089 | **0.565 ±0.065** | 0.537 ±0.088 |
| Cognitive task (7-way) | 0.686 ±0.042 | 0.796 ±0.052 | 0.727 ±0.058 | **0.879 ±0.042** | 0.869 ±0.062 | 0.783 ±0.078 | 0.783 ±0.063 | 0.740 ±0.060 | 0.885 ±0.038 | **0.900 ±0.044** | 0.888 ±0.049 | 0.859 ±0.063 | 0.887 ±0.042 | **0.893 ±0.051** | 0.876 ±0.058 | **0.895 ±0.054** |
| Cognitive task (24-way) | 0.237 ±0.022 | 0.373 ±0.019 | 0.315 ±0.026 | **0.469 ±0.030** | 0.452 ±0.030 | 0.322 ±0.024 | 0.308 ±0.028 | 0.243 ±0.029 | 0.459 ±0.031 | **0.479 ±0.031** | **0.469 ±0.026** | 0.427 ±0.018 | **0.469 ±0.037** | 0.462 ±0.031 | 0.456 ±0.033 | **0.459 ±0.025** |
| Autism diagnosis | 0.598 ±0.030 | 0.589 ±0.071 | 0.609 ±0.055 | 0.643 ±0.051 | **0.655 ±0.054** | 0.656 ±0.087 | 0.660 ±0.074 | 0.632 ±0.065 | 0.660 ±0.047 | **0.663 ±0.040** | 0.661 ±0.040 | 0.663 ±0.035 | **0.680 ±0.044** | 0.668 ±0.045 | 0.653 ±0.043 | **0.661 ±0.060** |
| AD diagnosis | 0.363 ±0.081 | 0.367 ±0.089 | 0.410 ±0.079 | **0.451 ±0.064** | 0.387 ±0.077 | 0.413 ±0.069 | 0.410 ±0.074 | 0.405 ±0.063 | 0.418 ±0.095 | **0.456 ±0.107** | **0.485 ±0.067** | 0.395 ±0.109 | 0.448 ±0.131 | 0.444 ±0.040 | 0.440 ±0.074 | **0.459 ±0.086** |
| FC stability | 0.696 ±0.085 | 0.677 ±0.068 | **0.729 ±0.044** | 0.643 ±0.054 | 0.650 ±0.045 | 0.682 ±0.045 | 0.681 ±0.045 | **0.689 ±0.053** | 0.635 ±0.047 | 0.644 ±0.043 | **0.620 ±0.046** | 0.612 ±0.044 | 0.615 ±0.043 | 0.609 ±0.045 | 0.598 ±0.045 | **0.603 ±0.044** |
| Fingerprinting | 0.069 ±0.083 | 0.230 ±0.166 | 0.424 ±0.206 | 0.682 ±0.217 | **0.696 ±0.201** | 0.570 ±0.216 | 0.576 ±0.221 | 0.527 ±0.220 | **0.796 ±0.194** | 0.776 ±0.172 | 0.856 ±0.156 | 0.863 ±0.150 | 0.852 ±0.164 | **0.875 ±0.151** | **0.886 ±0.145** | 0.884 ±0.148 |
| Age group classification | 0.260 ±0.047 | 0.386 ±0.077 | 0.413 ±0.093 | **0.455 ±0.120** | 0.452 ±0.136 | 0.411 ±0.054 | 0.402 ±0.059 | 0.328 ±0.059 | **0.478 ±0.105** | 0.473 ±0.048 | 0.480 ±0.112 | **0.515 ±0.075** | 0.433 ±0.079 | 0.497 ±0.104 | **0.477 ±0.096** | 0.475 ±0.096 |
| Crystallized intelligence | 0.376 ±0.040 | 0.454 ±0.072 | 0.490 ±0.075 | **0.530 ±0.078** | 0.472 ±0.095 | 0.501 ±0.086 | 0.507 ±0.089 | 0.483 ±0.082 | **0.526 ±0.097** | 0.505 ±0.082 | 0.497 ±0.100 | **0.542 ±0.092** | 0.516 ±0.117 | 0.525 ±0.101 | **0.516 ±0.102** | 0.515 ±0.114 |
| General intelligence | 0.396 ±0.080 | 0.418 ±0.064 | 0.442 ±0.079 | **0.469 ±0.098** | 0.442 ±0.104 | 0.445 ±0.102 | 0.448 ±0.094 | 0.439 ±0.082 | **0.467 ±0.104** | 0.461 ±0.108 | **0.463 ±0.098** | 0.417 ±0.099 | 0.446 ±0.085 | 0.458 ±0.121 | 0.428 ±0.122 | **0.459 ±0.112** |
| Autism cross-site | 0.560 ±0.118 | 0.608 ±0.069 | 0.620 ±0.113 | 0.640 ±0.067 | **0.662 ±0.068** | 0.679 ±0.083 | **0.680 ±0.080** | 0.662 ±0.112 | 0.662 ±0.086 | 0.635 ±0.091 | 0.667 ±0.061 | 0.655 ±0.092 | **0.696 ±0.136** | 0.640 ±0.110 | **0.638 ±0.144** | **0.638 ±0.166** |

## A.8 Effect of parcel number on downstream tasks

To examine whether downstream tasks prefer specific spatial resolutions, we varied the number of parcels for both DCA (41, 100, 200, 360, 400, 500) and Schaefer (100, 200, 300, 400, 500) atlases. For each resolution, we performed subject-level cross-validation and report accuracies in Tables 13 and 14. Across both atlas families, three consistent patterns emerge:

- **Peak-shaped (resolution-optimal) tasks.** Cognitive decoding shows a clear "rise-then-fall" profile: performance increases from coarse to intermediate resolutions and declines when parcels become excessively fine (e.g., DCA peaks around 200–360 parcels; Schaefer peaks typically at 300–400 for 7-way decoding), indicating an optimal meso-scale. Intuitively, coarse parcellations underfit task-relevant heterogeneity, whereas over-fragmentation reduces SNR per parcel, inflates inter-parcel edges, and amplifies misalignment across subjects, all of which hurt generalization.

- **Resolution-insensitive tasks.** Several clinical endpoints (e.g., AD diagnosis) exhibit weak or no monotonic trend across resolutions for both DCA and Schaefer. These tasks likely rely on subcortical or global signals that cortex-only atlases do not explicitly model, so varying cortical granularity alone has a limited effect. In such settings, choosing resolution can be guided by computational cost or downstream interpretability rather than accuracy.

- **Size-driven tasks.** Metrics whose value is mechanically affected by parcel size show systematic behavior: FC stability decreases beyond intermediate resolutions (consistent with lower within-parcel SNR and shorter time series per parcel), whereas subject fingerprinting improves with finer parcellations (more idiosyncratic, high-dimensional FC signatures), with the same tendencies observed for both atlas families.

Table 13: Downstream task performance across atlas resolutions for DCA

|  | DCA 41 | DCA 100 | DCA 200 | DCA 360 | DCA 400 | DCA 500 |
|---|---|---|---|---|---|---|
| Gender classification ↑ | 0.651 | 0.666 | 0.687 | **0.710** | 0.707 | 0.702 |
| Fluid intelligence ↑ | 0.429 | 0.491 | 0.497 | 0.535 | **0.543** | 0.537 |
| Cognitive task (7-way) ↑ | 0.842 | 0.869 | **0.900** | 0.887 | 0.882 | 0.895 |
| Cognitive task (24-way) ↑ | 0.426 | 0.452 | **0.479** | 0.469 | 0.465 | 0.459 |
| Autism diagnosis ↑ | 0.633 | 0.655 | 0.663 | **0.680** | 0.665 | 0.661 |
| AD diagnosis ↑ | 0.443 | 0.387 | 0.456 | 0.448 | 0.447 | **0.459** |
| FC stability ↑ | 0.642 | **0.650** | 0.644 | 0.615 | 0.609 | 0.603 |
| Fingerprinting ↑ | 0.435 | 0.696 | 0.776 | 0.852 | 0.811 | **0.884** |
| Age group classification ↑ | 0.408 | 0.452 | 0.473 | 0.433 | **0.512** | 0.475 |
| Crystallized intelligence ↑ | 0.521 | 0.472 | 0.505 | 0.516 | **0.523** | 0.515 |
| General intelligence ↑ | 0.439 | 0.442 | **0.461** | 0.446 | 0.448 | 0.459 |
| Autism cross-site ↑ | 0.636 | 0.662 | 0.635 | **0.696** | **0.696** | 0.638 |

Table 14: Downstream task performance across atlas resolutions for Schaefer

|  | Schaefer 100 | Schaefer 200 | Schaefer 300 | Schaefer 400 | Schaefer 500 |
|---|---|---|---|---|---|
| Gender classification ↑ | 0.628 | 0.668 | 0.670 | **0.726** | 0.694 |
| Fluid intelligence ↑ | 0.474 | 0.505 | 0.517 | 0.527 | **0.565** |
| Cognitive task (7-way) ↑ | 0.879 | 0.885 | 0.888 | **0.893** | 0.876 |
| Cognitive task (24-way) ↑ | **0.469** | 0.459 | 0.469 | 0.462 | 0.456 |
| Autism diagnosis ↑ | 0.643 | 0.660 | 0.661 | **0.668** | 0.653 |
| AD diagnosis ↑ | 0.451 | 0.418 | **0.485** | 0.444 | 0.440 |
| FC stability ↑ | **0.643** | 0.635 | 0.620 | 0.609 | 0.598 |
| Fingerprinting ↑ | 0.682 | 0.796 | 0.856 | 0.875 | **0.886** |
| Age group classification ↑ | 0.455 | 0.478 | 0.480 | **0.497** | 0.477 |
| Crystallized intelligence ↑ | **0.530** | 0.526 | 0.497 | 0.525 | 0.516 |
| General intelligence ↑ | **0.469** | 0.467 | 0.463 | 0.458 | 0.428 |
| Autism cross-site ↑ | 0.640 | 0.662 | **0.667** | 0.640 | 0.638 |

Taken together, there is no universally optimal parcel count. Intermediate resolutions (roughly 200–400 parcels) often strike a favorable trade-off for cortex-driven cognitive decoding, whereas

resolution-insensitive clinical tasks are robust across scales, and size-driven metrics move predictably with parcel granularity. These findings are consistent across DCA and Schaefer (Tables 13, 14) and provide practical guidance for selecting atlas resolution by task type rather than adopting a single fixed setting.

## A.9 Main task ablation

### A.9.1 Regularization and reconstruction loss

We investigated whether incorporating an orthogonality regularizer or a reconstruction loss would further improve parcellation quality (Table 15). Applying these augmented objectives to the same 100 subjects used in our main experiments yielded no statistically significant gains in homogeneity or silhouette coefficient for 100 parcels, indicating that the core KL-based clustering loss is sufficient to drive optimal voxel-level atlas generation.

**orthogonal loss** The orthogonality regularizer is defined on the centroid matrix $\mathbf{D} \in \mathbb{R}^{K \times d}$ (with unit-norm rows) as follows. Let

$$\mathbf{G} = \mathbf{D}\,\mathbf{D}^\top \quad (\in \mathbb{R}^{K \times K})$$

be the Gram matrix of pairwise inner products. We zero out the diagonal to isolate off-diagonal similarities:

$$\mathbf{G}_{\text{off}} = \mathbf{G} - \mathbf{I}_K.$$

The orthogonality loss then penalizes the mean absolute off-diagonal entry via

$$\mathcal{R}_\perp(\mathbf{D}) = \sqrt{\frac{1}{K(K-1)} \sum_{i \neq j} (\mathbf{G}_{\text{off}})_{ij}}.$$

Minimizing $\mathcal{R}_\perp$ encourages the rows of $\mathbf{D}$ to remain mutually orthogonal.

**masked reconstructed loss** We only consider the reconstruction of non-background voxels.

$$\mathcal{L}_{\text{masked\_MSE}} = \begin{cases} \dfrac{\sum_{i=1}^N (1-m_i)\left(\hat{x}_i - x_i\right)^2}{\sum_{i=1}^N (1-m_i)}, & \text{if } \sum_{i=1}^N (1-m_i) > 0, \\ 0, & \text{otherwise,} \end{cases}$$

where $\hat{x}_i$ and $x_i$ are the predicted and target values at voxel $i$, respectively, and $m_i \in \{0, 1\}$ is the binary mask indicating background ($m_i = 1$) or foreground ($m_i = 0$).

Table 15: Ablation study on orthogonality and masked reconstruction loss components.

| Loss | | | Homogeneity | Silhouette |
|---|---|---|---|---|
| KL | Orthogonality | Reconstruction | | |
| ✓ | | | 0.1002±0.0214 | 0.0301±0.0066 |
| ✓ | | ✓ | 0.0890±0.0091 | 0.0267±0.0045 |
| ✓ | ✓ | | **0.1004±0.0215** | **0.0304±0.0067** |

### A.9.2 Clustering on different smoothing level

To improve signal quality and spatial coherence, we applied spatial smoothing using AFNI's 3dBlurToFWHM [31], targeting a 3 mm full width at half maximum (FWHM). The preprocessed volumetric images were resampled to 2 mm isotropic resolution. This follows the common practice of setting the smoothing kernel to approximately 1.5 times the image resolution. For comparison, we also present results under two additional conditions: no smoothing and 6 mm FWHM smoothing. And there is no significant difference (Table 16).

Table 16: Evaluation of similarity metrics across smoothing levels and parcel resolutions.

| Metrics Raw | 100 | 200 | 360 | 500 | 800 |
|---|---|---|---|---|---|
| Homogeneity ↑ | 0.1004 | 0.1127 | 0.1266 | 0.1363 | 0.1535 |
| | ± 0.0216 | ± 0.0225 | ± 0.0229 | ± 0.0228 | ± 0.0226 |
| Silhouette ↑ | 0.0305 | 0.0422 | 0.0554 | 0.0647 | 0.0883 |
| | ± 0.0068 | ± 0.0094 | ± 0.0085 | ± 0.0090 | ± 0.0114 |

| Metrics FWHM 3mm | 100 | 200 | 360 | 500 | 800 |
|---|---|---|---|---|---|
| Homogeneity ↑ | 0.1004 | 0.1127 | 0.1266 | 0.1364 | 0.1536 |
| | ± 0.0216 | ± 0.0225 | ± 0.0230 | ± 0.0229 | ± 0.0227 |
| Silhouette ↑ | 0.0304 | 0.0417 | 0.0545 | 0.0644 | 0.0866 |
| | ± 0.0068 | ± 0.0078 | ± 0.0080 | ± 0.0086 | ± 0.0114 |

| Metrics FWHM 6mm | 100 | 200 | 360 | 500 | 800 |
|---|---|---|---|---|---|
| Homogeneity ↑ | 0.1005 | 0.1128 | 0.1267 | 0.1364 | 0.1536 |
| | ± 0.0217 | ± 0.0225 | ± 0.0229 | ± 0.0229 | ± 0.0227 |
| Silhouette ↑ | 0.0306 | 0.0419 | 0.0546 | 0.0642 | 0.0868 |
| | ± 0.0069 | ± 0.0078 | ± 0.0081 | ± 0.0083 | ± 0.0102 |

### A.9.3 Choice of graph cut method

In addition to spectral clustering, we evaluated several graph-cut algorithms (Fig. 10 and Table 17). However, most failed to guarantee that each resulting parcel forms a single connected subgraph, leading to fragmented regions. Here, we use a breadth-first search (BFS) based algorithm. The weighted BFS–connected clustering algorithm begins by converting the input edge list and weights into an undirected adjacency list, then randomly seeds $k$ initial clusters by assigning one unique node to each cluster. Each cluster maintains a max-heap of its unassigned neighboring nodes, prioritized by edge weight. Clusters then grow in parallel: at each step, a cluster pops the highest-weight neighbor from its heap, claims that node (if unassigned), and pushes all of its unassigned neighbors onto the heap. To enforce roughly equal cluster sizes, each cluster stops growing once it reaches $\lceil N/k \rceil$. If any nodes remain unassigned after this frontier-driven expansion, they are absorbed into the smallest adjacent cluster. By always selecting the strongest edges first and only adding connected nodes, this method produces contiguous clusters that respect the underlying graph's weighted connectivity.

Table 17: Evaluation of similarity metrics across graph cut methods and parcel resolutions.

| Metrics graph | 100 | 200 | 360 | 500 | 800 |
|---|---|---|---|---|---|
| Homogeneity ↑ | 0.1004 | 0.1127 | 0.1266 | 0.1364 | 0.1536 |
| | ± 0.0216 | ± 0.0225 | ± 0.0230 | ± 0.0229 | ± 0.0227 |
| Silhouette ↑ | 0.0304 | 0.0417 | 0.0545 | 0.0644 | 0.0866 |
| | ± 0.0068 | ± 0.0078 | ± 0.0080 | ± 0.0086 | ± 0.0114 |

| Metrics mst | 100 | 200 | 360 | 500 | 800 |
|---|---|---|---|---|---|
| Homogeneity ↑ | 0.0930 | 0.1026 | 0.1134 | 0.1210 | 0.1341 |
| | ± 0.0204 | ± 0.0210 | ± 0.0215 | ± 0.0219 | ± 0.0222 |
| Silhouette ↑ | 0.0223 | 0.0308 | 0.0407 | 0.0479 | 0.0607 |
| | ± 0.0052 | ± 0.0061 | ± 0.0065 | ± 0.0069 | ± 0.0072 |

### A.9.4 Choice of gray matter mask

In the main text, the corresponding ROI masks are extracted from FreeSurfer's aparc+aseg.mgz [48]. Here, we show the result (Table 18) by defining the ROI with the same mask to MMP [8]and Schaefer

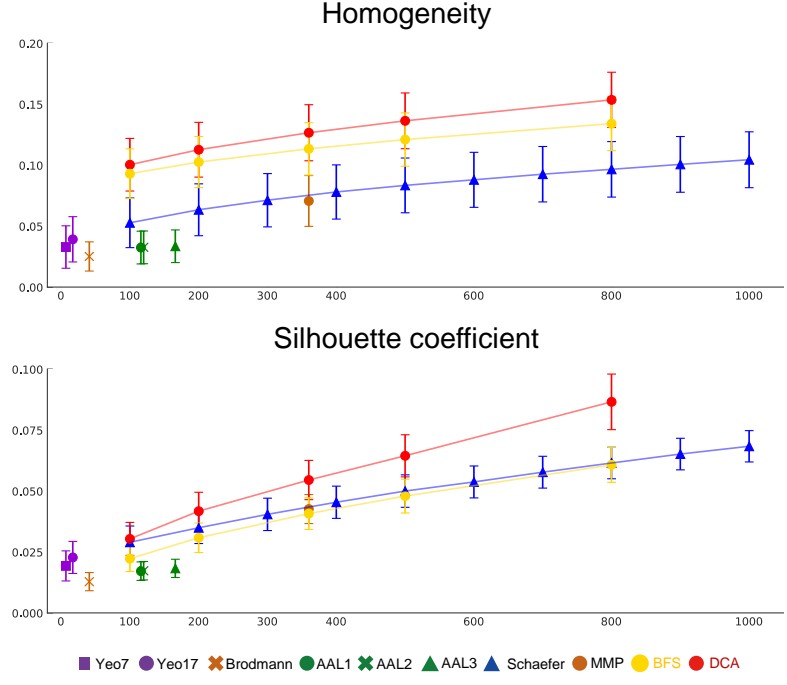

Figure 10: Homogeneity and Silhouette coefficients for weighted BFS–connected clustering and baselines, measured over 100 HCP subjects at varying numbers of parcels.

et al [2]. DCA still shows higher homogeneity and silhouette coefficients compared to corresponding atlas.

Table 18: Evaluation of Similarity Metrics on Schaefer, DCA, MMP, and DCA360 with Gray Matter Masks

|  | Schaefer100 | **DCA100** | MMP | **DCA360** |
|---|---|---|---|---|
| Homogeneity | 0.0885±0.0212 | **0.1004±0.0216** | 0.1212±0.0211 | **0.1266±0.0230** |
| Silhouette | 0.0191±0.0076 | **0.0304±0.0068** | 0.0470±0.0066 | **0.0545±0.0080** |

### A.9.5 Number of neighbours

We adopt the standard 26-neighbourhood in 3-D, which encompasses all voxels in a $3 \times 3 \times 3$ cube (excluding the centre) and thus captures both face- and diagonal interactions. To gauge its impact, we evaluated two reduced neighbourhoods—$K = 6$ (face-connected voxels only) and $K = 18$ (face + edge voxels)—alongside the full $K = 26$ setting. Table 19 shows that $K = 6$ and $K = 26$ yield nearly indistinguishable performance ($p > 0.05$, $t/U$-test), whereas $K = 18$ produces a noticeably lower silhouette score ($p < 0.05$). Overall, the full 26-neighbourhood provides stable and competitive results.

Table 19: Ablation on neighbourhood size.

| $K$ | **Homogeneity ↑** | **Silhouette ↑** |
|---|---|---|
| 6 | 0.1005±0.0216 | 0.0313±0.0071 |
| 18 | 0.1005±0.0310 | 0.0215±0.0068 |
| 26 | 0.1004±0.0215 | 0.0304±0.0067 |

### A.9.6 Distribution-based loss

We compared three common divergence objectives—Wasserstein distance, Jensen–Shannon divergence, and the KL divergence used in the main paper. Table 20 reveals virtually identical performance across all choices, with variations well within one standard deviation ( $p > 0.05$ ). This suggests that the framework is largely insensitive to the specific distribution-matching loss employed during clustering refinement.

Table 20: Reliability across distribution-based loss functions.

| Loss | Homogeneity ↑ | Silhouette ↑ |
|---|---|---|
| Wasserstein | 0.1003±0.0215 | 0.0306±0.0069 |
| JS divergence | 0.1005±0.0216 | 0.0308±0.0069 |
| KL divergence (used) | 0.1004±0.0216 | 0.0304±0.0068 |

### A.9.7 Centroid initialisation

Finally, we evaluated four initialisation schemes for the clustering centroids: *random*, *random+norm* (unit-normalised centroids), *xavier+norm*, and *orthogonal+norm*. As shown in Table 21, orthogonal+norm markedly lowers the first-epoch loss, indicating faster early convergence, yet all methods converge to nearly identical homogeneity and silhouette scores once training completes ( $p > 0.05$ ).

Table 21: Impact of centroid initialisation.

| Initialisation | 1st-epoch loss ↓ | Homogeneity ↑ | Silhouette ↑ |
|---|---|---|---|
| random | 6.5578±0.0774 | 0.1010±0.0218 | 0.0312±0.0070 |
| random+norm | 4.6250±0.0151 | 0.1004±0.0215 | 0.0307±0.0068 |
| xavier+norm | 4.6185±0.0207 | 0.1004±0.0215 | 0.0307±0.0067 |
| orthogonal+norm | **4.6062±0.0027** | 0.1004±0.0216 | 0.0304±0.0068 |

## A.10 Group atlas generation analysis

We systematically examined how three key hyperparameters in group-level atlas generation—voxel reliability threshold $\alpha$, template distinctiveness threshold $\beta$, and voxel inclusion threshold $\gamma$—affect downstream task performance. The following four tables report performance across 12 tasks using atlases of increasing resolution (100 to 500 parcels), under various parameter settings.

Table 22: Effect of group atlas generation parameters on performance (100 regions). Values are shown as mean ± standard deviation.

| $\alpha$ | 0.2 | | | | | | 0.3 | | | | | |
|---|---|---|---|---|---|---|---|---|---|---|---|---|
| $\beta$ | 0.2 | | | 0.3 | | | 0.2 | | | 0.3 | | |
| $\gamma$ | 0.7 | 0.8 | 0.9 | 0.7 | 0.8 | 0.9 | 0.7 | 0.8 | 0.9 | 0.7 | 0.8 | 0.9 |
| Gender classification | **0.703±0.063** | 0.666±0.080 | 0.648±0.074 | 0.676±0.066 | 0.674±0.050 | **0.684±0.057** | 0.640±0.067 | 0.627±0.067 | 0.626±0.066 | 0.637±0.082 | 0.657±0.089 | 0.651±0.079 |
| Fluid intelligence | 0.524±0.065 | 0.491±0.082 | 0.486±0.084 | 0.508±0.089 | 0.508±0.078 | 0.516±0.066 | 0.473±0.095 | 0.475±0.094 | 0.496±0.102 | **0.520±0.090** | 0.519±0.086 | 0.510±0.093 |
| Cognitive task (7-way) | 0.897±0.041 | **0.900±0.044** | 0.867±0.060 | 0.872±0.050 | 0.866±0.058 | 0.864±0.053 | 0.877±0.064 | 0.877±0.050 | **0.882±0.041** | 0.863±0.045 | 0.873±0.054 | 0.868±0.042 |
| Cognitive task (24-way) | **0.480±0.022** | 0.452±0.030 | 0.440±0.033 | 0.458±0.020 | 0.450±0.024 | 0.459±0.024 | 0.458±0.025 | 0.457±0.031 | 0.446±0.030 | 0.456±0.019 | 0.451±0.018 | 0.435±0.029 |
| Autism diagnosis | **0.672±0.057** | 0.655±0.054 | 0.653±0.024 | 0.658±0.032 | 0.658±0.026 | 0.665±0.046 | 0.652±0.042 | 0.668±0.044 | 0.667±0.021 | 0.634±0.047 | 0.647±0.040 | 0.642±0.039 |
| AD diagnosis | 0.448±0.119 | 0.387±0.077 | 0.387±0.099 | 0.432±0.064 | 0.428±0.073 | **0.451±0.054** | 0.428±0.080 | 0.443±0.057 | 0.417±0.064 | 0.424±0.083 | 0.447±0.070 | 0.448±0.105 |
| FC stability | 0.644±0.043 | 0.650±0.045 | 0.649±0.045 | 0.656±0.045 | 0.656±0.045 | 0.654±0.045 | 0.649±0.045 | 0.649±0.045 | 0.648±0.045 | **0.659±0.045** | **0.659±0.046** | **0.659±0.046** |
| Fingerprinting | **0.777±0.171** | 0.696±0.201 | 0.694±0.202 | 0.691±0.204 | 0.689±0.205 | 0.690±0.210 | **0.704±0.196** | 0.699±0.196 | 0.698±0.200 | 0.689±0.209 | 0.693±0.208 | 0.697±0.211 |
| Age group classification | **0.474±0.065** | **0.452±0.136** | 0.446±0.118 | 0.419±0.127 | 0.425±0.113 | 0.424±0.100 | 0.407±0.102 | 0.412±0.108 | 0.423±0.101 | 0.387±0.088 | 0.424±0.089 | 0.437±0.084 |
| Crystallized intelligence | 0.509±0.097 | 0.472±0.095 | 0.489±0.082 | 0.511±0.055 | 0.505±0.060 | **0.521±0.061** | 0.503±0.094 | 0.507±0.104 | 0.515±0.099 | 0.494±0.094 | 0.516±0.100 | 0.510±0.108 |
| General intelligence | 0.453±0.114 | 0.442±0.104 | 0.438±0.097 | 0.474±0.094 | 0.473±0.101 | **0.498±0.104** | 0.443±0.081 | 0.448±0.068 | 0.452±0.074 | 0.459±0.091 | 0.452±0.099 | 0.460±0.084 |
| Autism cross-site | 0.636±0.139 | 0.662±0.068 | 0.671±0.093 | 0.675±0.070 | **0.678±0.077** | 0.659±0.091 | 0.637±0.084 | 0.662±0.085 | 0.658±0.080 | 0.617±0.077 | 0.623±0.076 | 0.637±0.102 |

Table 23: Effect of group atlas generation parameters on performance (200 regions). Values are shown as mean ± standard deviation.

| $\alpha$ | 0.2 | | | | | | 0.3 | | | | | |
|---|---|---|---|---|---|---|---|---|---|---|---|---|
| $\beta$ | 0.2 | | | 0.3 | | | 0.2 | | | 0.3 | | |
| $\gamma$ | 0.7 | 0.8 | 0.9 | 0.7 | 0.8 | 0.9 | 0.7 | 0.8 | 0.9 | 0.7 | 0.8 | 0.9 |
| Gender classification | 0.682±0.062 | 0.686±0.054 | 0.692±0.054 | 0.678±0.042 | 0.682±0.044 | 0.686±0.043 | 0.698±0.042 | 0.686±0.057 | 0.677±0.056 | 0.678±0.037 | 0.691±0.105 | 0.677±0.056 |
| Fluid intelligence | 0.539±0.089 | 0.539±0.083 | **0.545±0.092** | 0.521±0.083 | 0.525±0.078 | 0.502±0.067 | 0.501±0.091 | **0.545±0.092** | 0.493±0.105 | 0.491±0.105 | 0.491±0.105 | 0.493±0.105 |
| Cognitive task (7-way) | 0.893±0.035 | 0.878±0.046 | 0.861±0.046 | 0.888±0.045 | 0.877±0.046 | 0.879±0.050 | 0.896±0.055 | 0.896±0.057 | 0.886±0.057 | 0.896±0.055 | 0.896±0.057 | 0.886±0.057 |
| Cognitive task (24-way) | 0.470±0.031 | 0.473±0.038 | 0.463±0.042 | 0.466±0.043 | 0.465±0.045 | 0.461±0.033 | 0.472±0.043 | 0.463±0.053 | 0.461±0.036 | 0.463±0.053 | 0.461±0.036 | 0.649±0.031 |
| Autism diagnosis | 0.655±0.058 | 0.647±0.048 | 0.664±0.068 | 0.657±0.036 | 0.650±0.035 | 0.646±0.046 | 0.651±0.063 | 0.647±0.064 | 0.664±0.062 | 0.646±0.062 | 0.432±0.058 | 0.649±0.031 |
| AD diagnosis | 0.430±0.126 | 0.426±0.132 | 0.449±0.144 | 0.378±0.148 | 0.407±0.124 | 0.407±0.132 | 0.430±0.144 | 0.447±0.064 | 0.449±0.144 | 0.407±0.132 | 0.432±0.058 | 0.450±0.082 |
| FC stability | 0.640±0.044 | 0.640±0.044 | 0.639±0.044 | 0.651±0.044 | 0.652±0.043 | 0.652±0.043 | **0.655±0.044** | 0.640±0.044 | 0.639±0.044 | 0.654±0.043 | **0.655±0.044** | 0.654±0.043 |
| Fingerprinting | 0.763±0.185 | 0.759±0.182 | 0.762±0.181 | 0.750±0.188 | 0.743±0.193 | 0.739±0.199 | 0.769±0.189 | 0.759±0.182 | 0.762±0.181 | 0.771±0.191 | 0.768±0.188 | 0.768±0.188 |
| Age group classification | 0.450±0.089 | 0.422±0.085 | 0.436±0.086 | 0.455±0.098 | 0.442±0.097 | 0.472±0.099 | 0.436±0.079 | 0.422±0.085 | 0.436±0.086 | 0.450±0.084 | 0.450±0.084 | 0.445±0.076 |
| Crystallized intelligence | 0.538±0.087 | **0.542±0.094** | 0.522±0.093 | 0.524±0.084 | 0.532±0.087 | 0.506±0.084 | 0.466±0.072 | **0.542±0.093** | 0.457±0.073 | 0.474±0.075 | 0.474±0.075 | 0.464±0.068 |
| General intelligence | **0.469±0.088** | 0.457±0.080 | 0.457±0.073 | 0.443±0.105 | 0.423±0.120 | 0.426±0.109 | 0.441±0.137 | 0.457±0.080 | 0.457±0.073 | 0.435±0.141 | 0.441±0.137 | 0.421±0.129 |
| Autism cross-site | **0.672±0.092** | 0.657±0.108 | 0.645±0.073 | 0.656±0.125 | 0.660±0.115 | 0.657±0.064 | 0.634±0.089 | 0.657±0.108 | 0.645±0.073 | 0.648±0.091 | 0.648±0.091 | 0.657±0.069 |

Table 24: Effect of group atlas generation parameters on performance (360 regions). Values are shown as mean ± standard deviation.

| α | 0.2 | | | | | | 0.3 | | | | | |
|---|---|---|---|---|---|---|---|---|---|---|---|---|
| β | 0.2 | | | 0.3 | | | 0.2 | | | 0.3 | | |
| γ | 0.7 | 0.8 | 0.9 | 0.7 | 0.8 | 0.9 | 0.7 | 0.8 | 0.9 | 0.7 | 0.8 | 0.9 |
| Gender classification | 0.705±0.050 | 0.710±0.059 | **0.715±0.064** | 0.692±0.064 | 0.701±0.060 | 0.681±0.054 | 0.685±0.055 | 0.683±0.058 | 0.683±0.049 | 0.688±0.094 | 0.682±0.084 | 0.691±0.087 |
| Fluid intelligence | 0.521±0.068 | 0.535±0.084 | 0.505±0.094 | 0.517±0.086 | 0.528±0.077 | 0.528±0.076 | 0.532±0.096 | 0.554±0.083 | 0.539±0.093 | 0.551±0.074 | 0.555±0.078 | **0.562±0.090** |
| Cognitive task (7-way) | 0.888±0.045 | 0.887±0.042 | 0.879±0.043 | 0.900±0.040 | 0.901±0.037 | 0.898±0.046 | 0.902±0.049 | **0.905±0.049** | 0.899±0.051 | 0.898±0.054 | 0.898±0.049 | 0.893±0.052 |
| Cognitive task (24-way) | 0.462±0.023 | 0.469±0.037 | 0.460±0.039 | 0.474±0.031 | 0.472±0.024 | 0.458±0.036 | 0.477±0.035 | 0.478±0.031 | 0.473±0.030 | **0.480±0.036** | 0.472±0.031 | 0.472±0.034 |
| Autism diagnosis | 0.673±0.043 | 0.680±0.044 | 0.658±0.058 | **0.688±0.025** | 0.677±0.056 | 0.685±0.055 | 0.654±0.049 | 0.658±0.058 | 0.653±0.046 | 0.660±0.065 | 0.661±0.050 | 0.670±0.059 |
| AD diagnosis | 0.463±0.118 | 0.448±0.131 | 0.463±0.117 | 0.458±0.083 | 0.462±0.080 | 0.432±0.082 | 0.432±0.121 | 0.402±0.119 | 0.452±0.113 | **0.474±0.133** | 0.459±0.110 | 0.455±0.104 |
| FC stability | 0.615±0.043 | 0.615±0.043 | 0.614±0.043 | 0.615±0.043 | 0.615±0.043 | 0.614±0.043 | 0.615±0.043 | 0.615±0.043 | 0.614±0.043 | 0.618±0.044 | **0.619±0.043** | 0.618±0.043 |
| Fingerprinting | 0.853±0.161 | 0.852±0.164 | 0.851±0.161 | 0.857±0.159 | 0.854±0.166 | 0.852±0.167 | 0.863±0.155 | 0.861±0.157 | 0.864±0.154 | **0.870±0.144** | 0.866±0.147 | 0.866±0.149 |
| Age group classification | 0.448±0.071 | 0.433±0.079 | 0.448±0.090 | 0.471±0.080 | **0.479±0.073** | 0.465±0.067 | 0.471±0.084 | 0.477±0.099 | 0.467±0.075 | 0.449±0.085 | 0.458±0.062 | 0.449±0.070 |
| Crystallized intelligence | 0.510±0.115 | 0.516±0.117 | 0.496±0.122 | 0.527±0.103 | **0.550±0.114** | 0.520±0.110 | 0.532±0.089 | 0.528±0.074 | 0.537±0.098 | 0.481±0.085 | 0.505±0.086 | 0.502±0.076 |
| General intelligence | 0.452±0.092 | 0.446±0.085 | 0.455±0.087 | 0.441±0.098 | 0.453±0.089 | 0.476±0.114 | 0.462±0.117 | **0.478±0.101** | 0.461±0.107 | 0.425±0.059 | 0.432±0.065 | 0.454±0.053 |
| Autism cross-site | 0.672±0.114 | **0.696±0.136** | 0.663±0.126 | 0.639±0.177 | 0.672±0.111 | 0.668±0.082 | 0.672±0.109 | 0.686±0.150 | 0.665±0.133 | 0.646±0.211 | 0.639±0.203 | 0.646±0.205 |

Table 25: Effect of group atlas generation parameters on performance (500 regions). Values are shown as mean ± standard deviation.

| α | 0.2 | | | | | | 0.3 | | | | | |
|---|---|---|---|---|---|---|---|---|---|---|---|---|
| β | 0.2 | | | 0.3 | | | 0.2 | | | 0.3 | | |
| γ | 0.7 | 0.8 | 0.9 | 0.7 | 0.8 | 0.9 | 0.7 | 0.8 | 0.9 | 0.7 | 0.8 | 0.9 |
| Gender classification | 0.711±0.081 | 0.702±0.056 | 0.707±0.056 | 0.708±0.057 | 0.710±0.057 | 0.719±0.041 | **0.742±0.056** | 0.728±0.067 | 0.714±0.069 | 0.703±0.051 | 0.702±0.060 | 0.700±0.062 |
| Fluid intelligence | 0.548±0.083 | 0.537±0.088 | 0.548±0.094 | 0.539±0.082 | 0.544±0.084 | **0.555±0.085** | 0.537±0.068 | 0.545±0.083 | 0.548±0.089 | 0.505±0.069 | 0.505±0.071 | 0.507±0.066 |
| Cognitive task (7-way) | 0.892±0.058 | 0.895±0.054 | 0.886±0.061 | 0.880±0.057 | 0.873±0.049 | 0.859±0.049 | **0.898±0.052** | 0.897±0.052 | 0.893±0.052 | 0.886±0.055 | 0.885±0.051 | 0.879±0.050 |
| Cognitive task (24-way) | 0.462±0.035 | 0.459±0.025 | 0.445±0.030 | 0.444±0.027 | 0.434±0.030 | 0.432±0.035 | **0.468±0.036** | **0.468±0.034** | 0.456±0.037 | 0.452±0.032 | 0.457±0.035 | 0.459±0.018 |
| Autism diagnosis | 0.649±0.070 | 0.661±0.060 | 0.647±0.047 | 0.633±0.089 | 0.590±0.051 | 0.635±0.050 | 0.655±0.059 | **0.667±0.040** | 0.640±0.063 | 0.575±0.070 | 0.580±0.109 | 0.572±0.085 |
| AD diagnosis | 0.448±0.092 | 0.459±0.086 | 0.455±0.094 | 0.455±0.095 | 0.429±0.092 | 0.474±0.093 | **0.504±0.105** | 0.489±0.113 | 0.497±0.114 | 0.448±0.101 | 0.471±0.107 | 0.478±0.077 |
| FC stability | 0.603±0.044 | 0.603±0.044 | 0.602±0.043 | 0.603±0.045 | 0.603±0.044 | 0.602±0.044 | 0.602±0.044 | 0.602±0.044 | 0.601±0.044 | **0.607±0.044** | **0.607±0.044** | **0.607±0.044** |
| Fingerprinting | 0.883±0.152 | 0.884±0.148 | 0.880±0.150 | 0.864±0.163 | 0.864±0.167 | 0.865±0.161 | **0.885±0.146** | **0.885±0.143** | 0.884±0.144 | 0.878±0.144 | 0.879±0.142 | 0.876±0.142 |
| Age group classification | 0.475±0.089 | 0.475±0.096 | **0.494±0.101** | 0.476±0.078 | 0.480±0.102 | 0.479±0.082 | 0.477±0.105 | 0.473±0.116 | 0.491±0.118 | 0.469±0.087 | 0.467±0.083 | 0.482±0.092 |
| Crystallized intelligence | 0.530±0.085 | 0.515±0.114 | 0.518±0.097 | 0.538±0.100 | 0.505±0.088 | 0.501±0.079 | 0.527±0.097 | 0.520±0.105 | 0.493±0.091 | **0.545±0.080** | 0.539±0.071 | 0.539±0.073 |
| General intelligence | 0.470±0.110 | 0.459±0.112 | 0.463±0.097 | 0.503±0.133 | 0.504±0.096 | **0.506±0.099** | 0.446±0.117 | 0.460±0.131 | 0.453±0.117 | 0.438±0.119 | 0.430±0.096 | 0.445±0.082 |
| Autism cross-site | 0.636±0.175 | 0.638±0.166 | **0.659±0.152** | 0.560±0.196 | 0.565±0.170 | 0.559±0.196 | 0.644±0.093 | 0.655±0.125 | 0.612±0.120 | 0.531±0.142 | 0.598±0.131 | 0.568±0.164 |

Our results reveal several trends. When the number of parcels is small (e.g., 100), using $\alpha = 0.2$ yields better performance, while $\alpha = 0.3$ becomes more advantageous as the resolution increases, possibly because finer parcellations require more voxels to capture individual variability—even if some voxels are less reliable. For $\beta$, a lower threshold ($\beta = 0.2$) tends to offer a marginal performance benefit, possibly due to increased distinctiveness between parcels. Although $\gamma = 0.7$ yields slightly better performance overall, we choose $\gamma = 0.8$ for group-level atlas generation to match the voxel coverage of MMP [8] and ensure comparability. Consequently, our final group atlas is generated using hyperparameters $\alpha = 0.2$, $\beta = 0.2$, and $\gamma = 0.8$.

### A.11 Spatiotemporal masking strategy

To ensure that masked regions form contiguous blocks in the high-resolution volume, we first spatially downsample the original $96 \times 96 \times 96$ grid by a factor of 16 along each axis, yielding a coarse $6 \times 6 \times 6$ volume (time dimension $T$ unchanged). We then sample two independent binary masks on this downsampled grid: a spatial mask with fraction $x_s$ of voxels set to zero, and a temporal mask with fraction $x_t$ of frames set to zero. We choose $x_s$ and $x_t$ so that the overall masking ratio satisfies

$$(1 - x_s)(1 - x_t) = 1 - r, \tag{10}$$

where $r$ is the desired fraction of masked spatiotemporal volume (e.g. $r = 0.8$ for 80% masking [57]). Finally, we upsample these binary masks back to the original resolution by expanding each down-sampled voxel mask to a $16^3$ block in space and each temporal mask entry to the corresponding contiguous frames. Applying the resulting mask to the full-resolution data yields large, continuous spatiotemporal occlusions, encouraging the encoder to reconstruct missing patches using both local and long-range context.

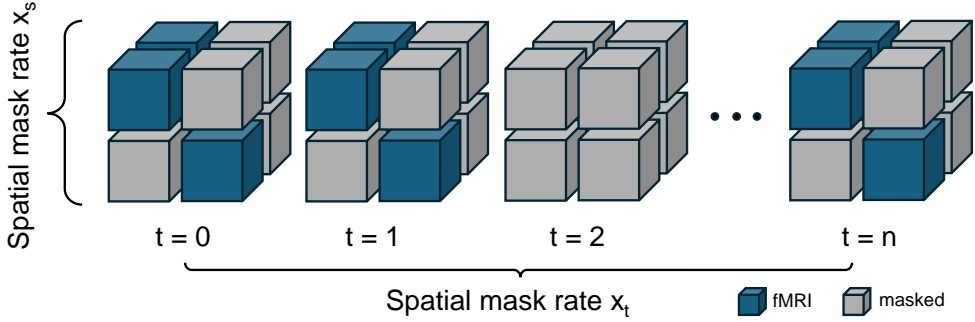

Figure 11: Illustration of the spatiotemporal masking scheme. We first select a fixed subset comprising $x_s$ of spatial voxels and mask them to zero across all time steps; additionally, we mask all voxels at a fraction $x_t$ of temporal frames to zero.

### A.12 Train and validation loss

We select the encoder checkpoint at epoch 8 and visualize the training losses, whose trajectories closely match those of other volumetric methods [58].

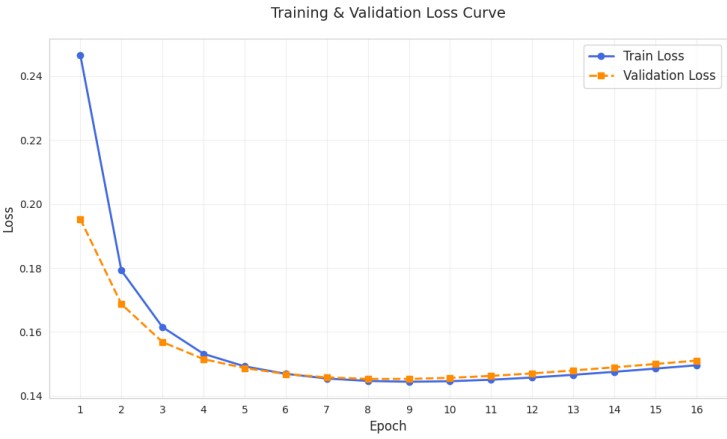

Figure 12: **Training losses.** We use the epoch-8 encoder checkpoint.

### A.13 Data organization

There are two parts in our repository, DCA and ATlaScore. Please follow `https://github.com/ncclab-sustech/DCA` for details.

#### A.13.1 DCA

The codes for pretraining and personalized clustering. One example subject is provided.

```
DCA/
├── data/
│   ├── fmri/
│   ├── mask/
│   ├── sub_test.txt
│   └── data_preparation.ipynb
├── results/
│   └── demo/
├── ablation_fmri.py
├── main.py
├── utils.py
└── swin_unetr.py
```

#### A.13.2 AtlaScore

The codes for evaluating atlases. The volumetric Homogeneity and Silhouette evaluations, as well as the surface-based DCBC evaluation, are provided. All 12 downstream tasks are implemented in `downstream/`, with FC features for DCA100 provided as an example.

```
AtlaScore/
├── similarity/
│   ├── compute_adj.py
│   ├── eva.py
│   └── eva_DCBC.py
├── downstream/
│   ├── docs/
│   ├── fc_data/
│   ├── nii_data/
│   ├── downstream.py
│   └── demo.ipynb
```

