# OpenReview forum: "DCA: Graph-Guided Deep Embedding Clustering for Brain Atlases"
_NeurIPS.cc/2025/Conference — NeurIPS 2025 poster_

### Official Review · Reviewer_zwAk · 2025-06-05

**Clarity:** 3
**Significance:** 2
**Originality:** 3
**Rating:** 5
**Confidence:** 3

**Summary:**

The paper suggests a method to derive personalized atlases for fMRI by learning the clustering jointly with finetuning the pretrained SWIN encoder backbone. Inspired by deep embedding clustering, clustering is learn by matching the learnt cluster centers with the results of spectral clustering of a 26-nearest neighbour graph derived from SWIN embedding outputs, where the adjacency is constructed with the cosine similarity between output embeddings.
To evaluate the atlases the paper uses homogeneity (averaged within parcel voxels correlation averaged across parcels later), silhouette score, and downstream tasks (eg age classification) performance.

**Questions:**

* Is there a conceptual reason not to replace Swin Transformer with Hiera [1] backbone, which is also hierarchical and is supposed to be simpler and better performing?
* In why in plot 3b there are no points for DCA below 100 to compare, for example to compare with Yeo 7 and Yeo 17 ?
* Do you have an explanation why homogeneity and silhouette score values are so low in general? How do you interpret it?
* Сould you provide the comparison of your predicted atlases with Schaefer et al [3] using intersection over union or similar metrics. It might be helpful to see how actually different are the atlases in terms of space and be more interpretable than homogeneity or silhouette score, especially for the people not familiar with the fMRI data. Are there any fully human labeled datasets for the atlases? Would [2] be helpful?
* In [3] they used metric close to homogeneity but also with accounting for the parcel size (see equation (2) from [3]). Why have you decided not to account for the parcel size here? Could we add the comparison between atlases using metric from [3]?
* In appendix Fig 4 and Table 5 - What is your explanation that DCA is worse with the DCBC metric than the other methods?


Minor
* In lines 129-130 you write *"In parallel, we build a 26-nearest-neighbour graph over the ROI masked voxels, weighting edges by pairwise embedding correlations. "*. Why 26? How sensitive is the network for the change of $k$ here (eg 20 instead of 26)? Does it come from some biological motivation or some previous research?
* In equation 2 (lines 163-164) you truncate the cosine similarity at 0. One could just keep $k$ edges with biggest absolute values. If its because inside the single region of interest everything should be only positively correlated, it might be worth adding an explicit explanation.

[1] Ryali, C., Hu, Y. T., Bolya, D., Wei, C., Fan, H., Huang, P. Y., ... & Feichtenhofer, C. (2023, July). Hiera: A hierarchical vision transformer without the bells-and-whistles. In International conference on machine learning (pp. 29441-29454). PMLR.
[2] Hermosillo, R. J., Moore, L. A., Feczko, E., Miranda-Domínguez, Ó., Pines, A., Dworetsky, A., ... & Fair, D. A. (2024). A precision functional atlas of personalized network topography and probabilities. Nature neuroscience, 27(5), 1000-1013.
[3] Schaefer, A., Kong, R., Gordon, E. M., Laumann, T. O., Zuo, X. N., Holmes, A. J., ... & Yeo, B. T. (2018). Local-global parcellation of the human cerebral cortex from intrinsic functional connectivity MRI. Cerebral cortex, 28(9), 3095-3114.

**Ethical Concerns:**

["NO or VERY MINOR ethics concerns only"]

**Final Justification:**

The authors have addressed my concerns during the review period, hence, I updated the score towards more positive

**Limitations:**

yes, though I would add a sentence that the number of parcels are hard fixed right now and the model does not have a capability to adjust it now

**Paper Formatting Concerns:**

no paper formatting concerns

**Quality:**

3

**Strengths And Weaknesses:**

Strengths
* DCA shows an improvement on homogeneity and silhouette score compared to the common atlases
* Adopting silhouette score for using correlations as a distance function is a neat idea as its not vulnerable for the arbitrary shapes of curse of dimensionality problems this way
* Paper provides the code and the plots contain error bars

Weaknesses
* DEC implies a specified number of clusters $k$ as input for the model, however, a discussion on what happens if $k$ is defined incorrectly is missing. [1-2] could be helpful for this. (varying $k$ could be interesting, especially if *"discover novel parcellations"*, lines 102-103, is the goal).
* The metric performance (homogeneity and silhouette score) seems to be poor for all of the methods, including the proposed DCA.
* Some crucial design choices (eg number of neighbours for the kNN graph for spectral clustering) are not explained and their impact is not experimentally explored

[1] Leiber, C., Strauß, N., Schubert, M., & Seidl, T. (2024). Dying Clusters Is All You Need--Deep Clustering With an Unknown Number of Clusters. arXiv preprint arXiv:2410.09491.
[2] Ronen, M., Finder, S. E., & Freifeld, O. (2022). Deepdpm: Deep clustering with an unknown number of clusters. In Proceedings of the IEEE/CVF Conference on Computer Vision and Pattern Recognition (pp. 9861-9870).

---

> ### Author Rebuttal · Authors · 2025-07-31
>
> > General response
>
> We thank all reviewers for their time and constructive feedback. We appreciate the recognition of the importance of our problem (Reviewers **puq2**, **uHxM**), the flexibility of our framework (Reviewers **o9NK**, **uHxM**), and the strengths of our method in both similarity metrics and downstream evaluations (Reviewers **puq2**, **o9NK**, **zwAk**).
>
> We also provide additional experiments to address reviewers' concerns on (1) model effectiveness by comparisons with recent state-of-the-art fMRI foundation model (Reviewer **zwAk**), (2) inadequate atlas comparison by additional atlas baselines (Reviewers **uHxM**), (3) downstream performance interpretation by new task evaluations and performance difference explanations (Reviewers **puq2**, **o9NK**), and (4) the impact of the number of parcels (Reviewers **o9NK**, **zwAk**).
>
> > Q1: Choice of Swin-UNETR
>
> We appreciate your insightful comments on the backbone selection. We chose `Swin-UNETR` as our encoder because it remains a widely adopted backbone for 3D medical image analysis, especially in volumetric neuroimaging tasks. To justify its suitability, we compared Swin-UNETR against two recent state-of-the-art foundation models [1, 2] using the full HCP and CHCP datasets. Possibly due to its ability to preserve dense voxel-level information, Swin-UNETR outperformed both models across two downstream tasks (**Rebuttal Table 1**).
>
> We agree that alternatives such as Hiera-based models (e.g., Medical-SAM2) represent promising and efficient backbones. Due to time constraints, we were unable to complete a systematic comparison of their training stability, computational cost, and downstream performance. We will include this discussion in the revised version.
>
> **Table 1: Backbone comparison: Classification for age and gender**
> ||Age accuracy ↑|Age F1 (macro) ↑|Gender accuracy ↑|Gender F1 (macro) ↑|
> |---|---|---|---|---|
> |Brain-JEPA [1]|0.419|0.151|0.663|0.657|
> |BrainLM [2]|0.373|0.179|0.528|0.485|
> |**Swin-UNETR**|**0.445**|**0.373**|**0.816**|**0.813**|
>
> > Q2: DCA 7/17/41 performance
>
> In our framework, DCA requires spatially contiguous ROIs, whereas Yeo 7 and 17 are functionally defined and spatially fragmented. Enforcing a very small number of parcels (e.g., 7 or 17) under spatial contiguity may compromise the functional interpretability of the resulting regions, which is why DCA 7/17 were not included in the main paper. To address the reviewer’s concern, we have now added these baselines (**Rebuttal Tables 2-3**).
>
> DCA still achieves high homogeneity at 7 and 17 parcels, but its silhouette scores are lower, likely because spatially adjacent yet functionally distinct regions are merged, while functionally similar but spatially distant regions (e.g., PCC and ACC in the DMN) are split into separate ROIs, reducing distinctiveness between ROIs. This spatial constraint also impacts downstream performance. For example, DCA 7/17 performs worse than Yeo in task decoding. In contrast, DCA 41 performs much better. In future work, relaxing the contiguity constraint from hard to soft may enable more meaningful network-level parcellation at extremely low resolutions.
>
> **Table 2: Evaluation of similarity metrics across atlases**
> ||Yeo 7|DCA 7|Yeo 17|DCA 17|Brodmann 41|DCA 41|
> |---|---|---|---|---|---|---|
> |Homogeneity ↑|0.0329±0.0174|0.0764±0.0186|0.0392±0.0186|0.0819±0.0193|0.0251±0.0120|**0.0892±0.0203**|
> |Silhouette ↑|0.0193±0.0062|0.0081±0.0038|**0.0228±0.0066**|0.0139±0.0045|0.0128±0.0037|0.0198±0.0053|
>
> **Table 3: Evaluation of downstream task performance across atlases**
> ||Yeo 7|DCA 7|Yeo 17|DCA 17|Brodmann 41|DCA 41|
> |---|---|---|---|---|---|---|
> |Gender classification ↑|0.547±0.077|0.619±0.052|0.620±0.063|**0.666±0.040**|0.659±0.078|0.651±0.045|
> |Fluid intelligence ↑|0.415±0.032|0.318±0.080|0.433±0.072|0.355±0.095|**0.456±0.046**|0.429±0.070|
> |Cognitive task (7-way) ↑|0.686±0.042|0.551±0.053|0.796±0.052|0.734±0.041|0.727±0.058|**0.842±0.060**|
> |Cognitive task (24-way) ↑|0.237±0.022|0.176±0.021|0.373±0.019|0.321±0.031|0.315±0.026|**0.426±0.046**|
> |Autism diagnosis ↑|0.598±0.030|0.617±0.051|0.589±0.071|0.617±0.040|0.609±0.055|**0.633±0.041**|
> |AD diagnosis ↑|0.363±0.081|0.371±0.087|0.367±0.089|**0.443±0.096**|0.410±0.079|**0.443±0.081**|
> |FC stability ↑|0.696±0.085|0.631±0.104|0.677±0.068|0.696±0.054|**0.729±0.044**|0.642±0.052|
> |Fingerprinting ↑|0.069±0.083|0.134±0.129|0.230±0.166|0.218±0.149|0.424±0.206|**0.435±0.220**|
> |Age group classification ↑|0.260±0.047|0.284±0.049|0.386±0.077|0.363±0.069|**0.413±0.093**|0.408±0.101|
> |Crystallized intelligence ↑|0.376±0.040|0.431±0.069|0.454±0.072|0.428±0.068|0.490±0.075|**0.521±0.069**|
> |General intelligence ↑|0.396±0.080|0.350±0.068|0.418±0.064|0.407±0.071|**0.442±0.079**|0.439±0.070|
> |Autism cross-site ↑|0.560±0.118|0.600±0.076|0.608±0.069|0.617±0.095|0.620±0.113|**0.636±0.083**|
>
> > Q3: Homogeneity and silhouette performance interpretation
>
> We thank the reviewer for raising this point. While the absolute values of homogeneity and silhouette scores may appear low, their overall scale is comparable to those reported in previous work [3]. Minor differences are likely due to variations in preprocessing pipelines, especially ICA-based denoising and spatial smoothing, which can affect the absolute magnitude of these metrics. Moreover, evaluation results can also differ depending on whether the atlas is assessed in the volume space or on the surface representation of the brain. With other conditions controlled, the relative ranking of atlases remains consistent, and DCA consistently outperforms baselines across resolutions (**Appendix Tables 3-4**).
>
> > Q4: Comparison with Schaefer atlases
>
> To assess spatial overlap, we compared `DCA` and `Schaefer` atlases at 100, 200, and 500 parcels using Hungarian matching. It obtained mean Dice scores of 0.443, 0.433, and 0.427, and corresponding IoU scores of 0.307, 0.302, and 0.293. Network-specific analysis showed higher overlap in the visual network (Dice = 0.489-0.531; IoU = 0.338-0.380), and lower overlap in the default mode network (Dice = 0.421–0.446; IoU = 0.290–0.304), suggesting that DCA preserves structure in sensory regions and refines higher-order areas.
>
> While reference [4] is highly relevant, its volume-space data is not yet publicly available. We therefore used the Brodmann atlas, one of the most widely adopted manually labeled volumetric references, for anatomical comparison. We computed normalized mutual information (NMI) with the Brodmann atlas, which does not require label matching. `DCA` consistently achieved slightly higher NMI than Schaefer (e.g., 0.592 vs. 0.580 at 100 parcels), indicating better preservation of macro-anatomical organization.
>
> > Q5: Homogeneity metric definition
>
> We apologize for the oversight. The main paper only presented the simplified form of the global homogeneity metric. In practice, we correctly account for parcel size variations following Equation (2) from [5] (see **Code lines 84–90** in AtlaScore/similarity/eva.py). We will revise it.
>
> > Q6: DCBC metric performance interpretation
>
> While DCBC is a surface-based metric, our DCA atlases are constructed in volumetric space. To compute DCBC, we projected both fMRI data and atlas labels onto the fsLR-32k surface using the neuromaps toolbox [6]. This transformation may introduce coordinate discrepancies, especially in regions with complex cortical folding, and reduce the performance of voxel-based atlases like DCA. This could partly explain the lower DCBC scores. We included this metric to enable comprehensive comparisons with prior surface-based methods.
>
> > Q7: Choice of number of neighbors
>
> We apologize for the lack of clarity. We adopt the standard 26-neighborhood in 3D, which includes all voxels in a 3×3×3 cube excluding the center. It captures both adjacent and diagonal interactions. We additionally tested with 6 and 18 neighbors, and observed similar performance ($p>0.05$. See **Rebuttal Table 3** in response to Reviewer **uHxM**).
>
> > Q8: Cosine similarity truncation
>
> We appreciate your thoughtful observation. While our earlier implementation required graph positive semi-definiteness, the current version (see **line 123** in DCA/utils.py) no longer truncates cosine similarity. We regret the documentation oversight and will ensure all methods are accurately described in the revision.
>
> > Weakness 1: Choice of number of parcels $K$
>
> When $K$ is small, heterogeneous functional areas tend to be merged into a single parcel. Increasing $K$ usually improves accuracy (**Rebuttal Tables 2-3**), but may fragment a single functional region into multiple parcels and substantially raises computational cost.
>
> Varying $K$ provides insights into how different resolutions capture task-relevant structure (**Rebuttal Tables 2-3** in response to Reviewer **o9NK**). While an optimal $K$ is task-dependent and may not exist in a task-agnostic setting, certain resolutions may better suit specific applications. Our framework naturally supports multiresolution parcellations and can be extended with automatic-$K$ clustering and task-supervised objectives to generate individualized, performance-optimized atlases.
>
> ---
>
> Reference:
>
> [1] Brain-JEPA: Brain Dynamics Foundation Model with Gradient Positioning and Spatiotemporal Masking. NeurIPS 2024.
>
> [2] BrainLM: A foundation model for brain activity recordings. ICLR 2024.
>
> [3] Atlas-guided parcellation: Individualized functionally-homogenous parcellation in cerebral cortex. Computers in Biology and Medicine 2022.
>
> [4] A precision functional atlas of personalized network topography and probabilities. Nature Neuroscience 2024.
>
> [5] Local-global parcellation of the human cerebral cortex from intrinsic functional connectivity MRI. Cerebral Cortex 2018.
>
> [6] Neuromaps: structural and functional interpretation of brain maps. Nature Methods 2022.

---

> > ### Comment · Reviewer_zwAk · 2025-08-01
> > **Thanks for the reply**
> >
> > Thank you for the clear answer and thoughtfully addressing all the raised comments.
> >
> > One question I still have about **Q3** - I do not question that your homogeneity and silhouette performance are comparable with and outperform the previous baselines. I was more wondering how exactly you interpret it -  e.g. do they have any fundamental limitations (e.g. tend to be too low in with high-dimensional data -  are these are actually the best metrics to be used?), or is it just all the task is so hard that all of the current methods perform poorly on it?

---

> ### Author Response · Authors · 2025-08-01
>
> We appreciate your insightful question regarding the interpretation and limitations of internal clustering metrics. Indeed, `homogeneity` and `silhouette` scores can be affected by the high dimensionality and intrinsic noise of voxel-level fMRI data. Moreover, increasing the number of parcels ($K$) would inevitablely increase homogeneity score, due to the inherently smooth nature of fMRI signals. As a result, comparing homogeneity scores across different parcellation scales can be misleading, and internal metrics alone are insufficient for comprehensive evaluation. In addition, the requirement for spatial contiguity further constrains cluster boundaries, making it more challenging to achieve very high performance on these metrics.
>
> Nonetheless, these measures remain widely used and meaningful for atlas evaluation. Homogeneity directly reflects within-parcel signal similarity, while the silhouette coefficient captures the tradeoff between within- and between-parcel separability—both of which align with the core goal of functionally coherent and spatially distinct brain regions [1, 2]. Importantly, in DCA, we do not optimize these two metrics directly; instead, they emerge from unsupervised clustering in latent space, which underscores the quality of the learned representations.
>
> While task-specific metrics (e.g., decoding accuracy or diagnostic performance) can provide additional insight, they often favor atlases tuned to particular applications or populations. In contrast, homogeneity and silhouette offer task-agnostic, interpretable measures for evaluating general-purpose atlases. Therefore, although the absolute values may appear modest due to the complexity of the problem, the consistent relative improvements across methods and resolutions reflect meaningful progress in this challenging setting.
>
> [1] Atlas-guided parcellation: Individualized functionally-homogenous parcellation in cerebral cortex. Computers in Biology and Medicine 2022
>
> [2] Local-Global Parcellation of the Human Cerebral Cortex from Intrinsic Functional Connectivity MRI. Cerebral Cortex. 2018

---

> ### Author Response · Authors · 2025-08-04
> **Task-specific metrics and model adaptation**
>
> We demonstrate that our framework can be readily adapted to task-specific settings, achieving substantial improvements on the corresponding evaluation metrics. Specifically, we replace the reconstruction self-supervised Swin-UNETR with a version fine-tuned for gender classification (Table 1) and use its encoder to derive a task-specific atlas, denoted $\text{DCA}_{100}^{\mathrm{gender}}$. Using this atlas to aggregate fMRI signals into 100 ROIs for $N=200$ subjects drawn from the Swin UNETR fine-tuning *test* split to avoid data leakage, we train a compact neural network, two 1D convolutional layers followed by two fully connected layers, on the ROI-level time series (identical to the setup in Weakness 1 / Table 3 to Reviewer puq2) and evaluate on 20% held-out. Accuracy increases to **82%** (**+12%** over original method).
>
> These results indicate that targeted encoder fine-tuning yields task-specific atlases that deliver measurable gains on the target task, without compromising the spatial continuity of the atlas.
>
> **Table 4: Gender Classification Results**
> |Atlas|Accuracy ↑|F1 (Macro) ↑|F1 (Weighted) ↑|
> |---|---|---|---|
> |Watershed (100)|0.73|0.73|0.73|
> |Schaefer100|0.65|0.65|0.65|
> |DCA100 (group)|0.70|0.70|0.70|
> |DCA100 (individual)|0.70|0.69|0.70|
> |DCA$_{100}^{\text{gender}}$ (group)|0.70|0.67|0.67|
> |DCA$_{100}^{\text{gender}}$ (individual)|**0.82**|**0.82**|**0.82**|

---

> > ### Comment · Reviewer_zwAk · 2025-08-05
> > **Thanks**
> >
> > Thank you for your elaborate explanations and additional experiments.
> > I have no further questions for now, but will follow the discussions with other reviewers.

---

> ### Author Response · Authors · 2025-08-07
> **Thank you for the review**
>
> Thank you once again for your valuable feedback—especially your remarks on evaluation metrics inspired us to explore atlas fine-tuning, which has strengthened the paper. We are now in the final stage of the rebuttal process and have addressed each reviewer’s concerns point by point, receiving positive responses.
>
> Could we kindly invite you to take a brief look at the latest discussion thread? If you have any additional comments or questions, please let us know at your earliest convenience so we have sufficient time to incorporate them. We hope the new material will also earn your positive assessment.

---

### Official Review · Reviewer_uHxM · 2025-06-19

**Clarity:** 2
**Significance:** 3
**Originality:** 3
**Rating:** 4
**Confidence:** 3

**Summary:**

This paper proposes Deep Cluster Atlas (DCA), a novel deep learning framework for generating individualized, voxel-wise brain atlases from resting-state fMRI data. Unlike traditional atlases that are predefined and group-level, DCA uses graph-guided deep embedding clustering to create functionally coherent and spatially contiguous brain parcellations with controllable resolution and anatomical coverage. The method supports both group-level and subject-specific atlas construction and generalizes across different brain regions.

**Questions:**

1. I noticed that the Related Work section includes several recent approaches on MRI-based brain atlas construction and brain segmentation published in the past two to three years. However, these methods are not included in the comparative experiments. Could the authors clarify whether this omission is due to the lack of publicly available implementations or other reasons?
2. In addition to the KL divergence loss, it would be interesting to know whether the authors have experimented with other distribution-based loss functions, and how those choices might impact the model's performance.

**Ethical Concerns:**

["NO or VERY MINOR ethics concerns only"]

**Limitations:**

Please see the weakness.

**Paper Formatting Concerns:**

No paper formatting issues.

**Quality:**

3

**Strengths And Weaknesses:**

Strengths:
1. The paper identifies critical limitations of existing brain atlases: fixed resolution, group-level averaging, and lack of flexibility and individual specificity, convincingly argues for the need for individualized, voxel-level, and flexible parcellation methods that better reflect both functional and anatomical information.
2. The proposed DCA framework has good flexibility and generalizability; the standardized benchmarking platform provides quantitative evidence of DCA's superiority over several state-of-the-art atlases across a range of tasks and spatial scales.
3. The author provided the source code and corresponding results.

Weakness:
1. The comparative methods included in the main experiments are relatively outdated, lacking any recent approaches proposed after 2020, which may weaken the relevance and rigor of the evaluation.
2. The figures in the main text suffer from low resolution (the appendix is good), and enhancing their clarity would significantly improve the readability and presentation quality of the paper.
3. The ablation analysis in the main paper lacks depth, focusing solely on coarse-grained, modular ablations without exploring finer-grained factors that could provide deeper insights into the contribution of each component.

---

> ### Author Rebuttal · Authors · 2025-07-31
>
> > General response
>
> We thank all reviewers for their time and constructive feedback. We appreciate the recognition of the importance of our problem (Reviewers **puq2**, **uHxM**), the flexibility of our framework (Reviewers **o9NK**, **uHxM**), and the strengths of our method in both similarity metrics and downstream evaluations (Reviewers **puq2**, **o9NK**, **zwAk**).
>
> We also provide additional experiments to address reviewers' concerns on (1) model effectiveness by comparisons with recent state-of-the-art fMRI foundation model (Reviewer **zwAk**), (2) inadequate atlas comparison by additional atlas baselines (Reviewers **uHxM**), (3) downstream performance interpretation by new task evaluations and performance difference explanations (Reviewers **puq2**, **o9NK**), and (4) the impact of the number of parcels (Reviewers **o9NK**, **zwAk**).
>
> > Weakness 1: Outdated comparative baselines
>
> Thank you for your valuable suggestion. In the main paper, our comparative baselines were selected based on popularity and historical influence, including widely used atlases such as Yeo, Brodmann, Schaefer, and MMP. We acknowledge that this focus on classical atlases may have excluded more recent methodological advances, potentially limiting the completeness of our comparison.
>
> Recent approaches in our Related Work generate probabilistic maps or segment only specific regions, without offering whole-brain or cortex-wide deterministic atlases directly comparable to ours [1, 2]. We recognize the importance of including recent atlases for a fair comparison and will add more up-to-date atlases to Related Work. To address this concern, we added comparisons with `GIANT`, `Watershed `, `Allen`, and `MUSE`.
> Specifically, we incorporated three representative recent atlases from [3]. We also added the Allen Human Reference Atlas - 3D, 2020. These additions improve the breadth and contemporaneity of our comparisons (**Rebuttal Table 1-2**). Our approach still demonstrates overall better performance. While the new atlas shows lower scores on certain evaluation metrics, it should be noted that these atlases are whole-brain rather than cortex-only, which offers clear advantages for tasks that likely involve subcortical features. In the supplementary material, we also provide a non-cortical version of the atlas. We believe that integrating our DCA atlas with subcortical regions could further enhance its utility.
>
> **Table 1: Evaluation of similarity metrics across atlases at low (≤ 100 parcels) and medium (> 100 parcels) resolution**
> ||GIANT 50|Watershed 100|DCA 100|Allen 141|MUSE 149|DCA 200|
> |---|---|---|---|---|---|---|
> |Homogeneity ↑|0.0148±0.0073|0.0143±0.0070|**0.1004±0.0216**|0.0230±0.0106|0.0208±0.0080|**0.1127±0.0225**|
> |Silhouette ↑|0.0079±0.0023|0.0078±0.0020|**0.0304±0.0068**|0.0164±0.0055|0.0149±0.0035|**0.0417±0.0078**|
>
> **Table 2: Evaluation of downstream task performance across atlases at low (≤ 100 parcels) and medium (> 100 parcels) resolution**
> ||GIANT 50|Watershed 100|DCA 100|Allen 141|MUSE 149|DCA 200|
> |---|---|---|---|---|---|---|
> |Gender classification ↑|0.654±0.073|0.646±0.052|**0.666±0.080**|0.645±0.109|0.681±0.041|**0.687±0.073**|
> |Fluid intelligence ↑|0.384±0.098|0.425±0.053|**0.491±0.082**|0.434±0.095|0.456±0.071|**0.497±0.074**|
> |Cognitive task (7-way) ↑|0.670±0.065|0.383±0.050|**0.869±0.062**|0.622±0.053|0.749±0.038|**0.900±0.044**|
> |Cognitive task (24-way) ↑|0.266±0.024|0.104±0.013|**0.452±0.030**|0.171±0.018|0.261±0.030|**0.479±0.031**|
> |Autism diagnosis ↑|0.610±0.038|0.561±0.046|**0.655±0.054**|0.646±0.062|0.636±0.057|**0.663±0.040**|
> |AD diagnosis ↑|**0.459±0.064**|0.403±0.083|0.387±0.077|**0.467±0.090**|0.425±0.085|0.456±0.107|
> |FC stability ↑|**0.733±0.048**|0.731±0.053|0.650±0.045|0.664±0.066|**0.684±0.058**|0.644±0.043|
> |Fingerprinting ↑|0.358±0.205|0.347±0.216|**0.696±0.201**|0.493±0.248|0.539±0.256|**0.776±0.172**|
> |Age group classification ↑|0.343±0.047|0.349±0.058|**0.452±0.136**|0.394±0.072|0.353±0.051|**0.473±0.048**|
> |Crystallized intelligence ↑|0.462±0.089|0.447±0.076|**0.472±0.095**|0.488±0.046|**0.505±0.069**|**0.505±0.082**|
> |General intelligence ↑|0.381±0.091|0.406±0.082|**0.442±0.104**|0.375±0.050|**0.465±0.116**|0.461±0.108|
> |Autism cross-site ↑|0.611±0.066|0.568±0.114|**0.662±0.068**|0.595±0.166|0.609±0.107|**0.635±0.091**|
>
> > Weakness 2: Low-resolution figures in main text
>
> Thank you for pointing this out. We will improve the resolution of the figures in the main text to enhance clarity and overall presentation quality.
>
> > Weakness 3: Coarse ablation and Loss analysis
>
> Due to space limitations, we provide additional ablations and discussions in the supplementary material, covering multiple aspects of our framework. We will also include a more detailed discussion of the ablation results in the main paper in the revised version.
>
> 1. Model architecture – removing key components degrades performance and breaks spatial continuity (**Appendix 6.1**);
>
> 2. Regularization options and SimMIM loss – adding these has a negligible impact on final metrics (**Appendix 6.2**);
>
> 3. Pretraining data preprocessing – different preprocessing strategies do not affect model performance (**Appendix 6.3**);
>
> 4. Data smoothing – all smoothed variants evaluated with DCA outperform baseline atlases (**Appendix 5.1**);
>
> 5. Graph segmentation methods – spectral clustering consistently outperforms alternative approaches (**Appendix 6.4**);
>
> 6. Tissue segmentation templates – results remain superior to baselines regardless of template choice (**Appendix 6.5**);
>
> 7. Hyperparameters for group-level atlas generation – Impact of different hyperparameters on downstream tasks (**Appendix 7**).
>
> And following the reviewers’ suggestions, we have now included additional ablation studies to further validate our design choices.
>
> **(1) Number of Neighbors.** We adopt the standard 26-neighborhood in 3D, which includes all voxels in a 3×3×3 cube excluding the center voxel, thereby capturing both adjacent and diagonal spatial interactions. To analyze its effect, we further tested different neighborhood sizes ($K=6,18,26$), corresponding respectively to: (i) only face-connected voxels, (ii) face+edge connections, and (iii) all voxels in the 3×3×3 cube excluding the center.
>
> As shown in **Table 3**, the results for $K=6$ and $K=26$ are very close ($p>0.05$, t-test/U-test), while using $K=18$ yields a noticeably lower silhouette score ($p<0.05$, U-test). Overall, the choice of $K=26$ provides stable and competitive performance.
>
> **Table 3: Ablation study on neighborhood size selection**
> ||Homogeneity ↑|Silhouette ↑|
> |---|---|---|
> |6|0.1005±0.0216|0.0313±0.0071|
> |18|0.1005±0.0310|0.0215±0.0068|
> |26|0.1004±0.0215|0.0304±0.0067|
>
> **(2) Distribution-based Loss.** We added additional experiments to compare 3 commonly used distribution-based loss functions for atlas generation (**Table 4** ). We observe that Wasserstein, JS, and KL divergence losses yield nearly identical performance on both homogeneity and silhouette metrics, with only negligible variations within the standard deviation range ($p>0.05$, t-test/U-test). This suggests that our framework is largely insensitive to the specific choice of distribution-based loss, and all 3 objectives are equally effective in guiding the clustering refinement process.
>
> **Table 4: Reliability on distribution-based loss functions**
> ||Homogeneity ↑|Silhouette ↑|
> |---|---|---|
> |Wasserstein Loss|0.1003±0.0215|0.0306±0.0069|
> |JS Divergence|0.1005±0.0216|0.0308±0.0069|
> |KL Divergence (used)|0.1004±0.0216|0.0304±0.0068|
>
> **(3) Initialization.** We compared three alternative centroid initialization strategies for embedding clustering (`random`, `random+norm`, `xavier+norm`, and `orthogonal+norm` in **Table 5**). Using our `orthogonal+norm` initialization accelerates convergence in the early training stage, while its impact on the final performance metrics is negligible once the model has fully converged ($p>0.05$, t-test/U-test).
>
> **Table 5: Comparisons on different initialization methods**
> ||First epoch loss ↓|Homogeneity ↑|Silhouette ↑|
> |---|---|---|---|
> |random|6.5578±0.0774|0.1010±0.0218|0.0312±0.0070|
> |random+norm|4.6205±0.0151|0.1004±0.0215|0.0307±0.0068|
> |xavier+norm|4.6185±0.0207|0.1004±0.0215|0.0307±0.0067|
> |orthogonal+norm (used)|**4.6062±0.0027**|0.1004±0.0216|0.0304±0.0068|
>
> ---
>
> Reference:
>
> [1] A hierarchical Bayesian brain parcellation framework for fusion of functional imaging datasets. Imaging Neuroscience 2025.
>
> [2] A hierarchical atlas of the human cerebellum for functional precision mapping. Nature Communications 2024.
>
> [3] A genetically informed brain atlas for enhancing brain imaging genomics. Nature Communications 2025.

---

> > ### Comment · Reviewer_uHxM · 2025-08-06
> > **Reply to authors.**
> >
> > Thank you for the authors’ response and the additional experiments. The response has addressed my concerns. I hope the authors can adjust the paper’s length allocation in the following version, moving some important ablation studies (e.g., Appendix 6.1) into the main text and improving the quality of the figures. Overall, I believe this paper is close to the NeurIPS acceptance standard, and I will maintain my borderline accept score.

---

> > > ### Author Response · Authors · 2025-08-06
> > >
> > > We sincerely appreciate the reviewer's recognition of our work.  In the subsequent version, we will improve the image resolution and supplement key findings in the main text. We believe that in this era of large models, novel approaches are needed to revitalize the atlas field. We sincerely appreciate all your feedback and suggestions.

---

> ### Author Response · Authors · 2025-08-05
> **Supplement to Question 1**
>
> We would like to briefly supplement your **Question 1**, which referenced “atlas construction” and “brain segmentation.”
> For **(1) new atlas baselines**, we have already explained this in our prior response and added four recent, publicly available atlases as baselines. Our method performs better on these comparisons on similarity metrics (Table 1), downstream task performance for group atlas (Table 2) and personalized atlas (Table 3-4 to Reviewer puq2).
>
> Here we clarify why we do not include **(2) MRI segmentation baselines**. Beyond using fMRI rather than MRI, the key point is that our approach is fully self-supervised, whereas brain segmentation typically requires labels. Training with predefined label templates turns the task into atlas **registration** rather than atlas **generation**—for example, DDparcel trains with an atlas template, allowing the atlas to be quickly registered to new subjects [1]. In contrast, our method aims to generate new, personalized atlases. The core innovation is a self-supervised framework that yields spatially continuous atlases without any voxel-level labels.
>
> Moreover, our self-supervised framework can be augmented with fine-tuning to boost task-specific performance. We first fine-tune the encoder on an **external dataset** with individual labels (e.g., gender) using a supervised classification objective, and then use this encoder to generate atlases in the same unsupervised manner. Without using any additional information from the evaluation subjects, this yields task-specific atlases and substantially improves classification on the corresponding task (**+10%** with the graph-based method and **+12%** with the CNN classifier; Tables 6–7 to Reviewer puq2).
>
>
> [1] DDParcel: Deep Learning Anatomical Brain Parcellation From Diffusion MRI. IEEE Transactions on Medical Imaging 2024.

---

### Official Review · Reviewer_o9NK · 2025-06-27

**Clarity:** 3
**Significance:** 3
**Originality:** 3
**Rating:** 5
**Confidence:** 5

**Summary:**

This paper introduces Deep Cluster Atlas (DCA), a novel framework for generating individualized and group-level voxel-wise brain parcellations. DCA leverages a pretrained Swin-UNETR encoder combined with spatially regularized deep clustering to create high-resolution brain atlases that are both functionally coherent and anatomically contiguous. It overcomes limitations of traditional brain atlases, which are often fixed, group-level templates with limited flexibility. DCA allows flexible control over parcellation granularity and anatomical scope, and it outperforms existing methods in terms of homogeneity, silhouette coefficient, and performance on downstream tasks like autism diagnosis and cognitive decoding.

**Questions:**

1. Regarding the Voxel-level atlas evaluation platform, how is functional connectivity derived? Is it different from the standard method, which computes the Pearson correlation coefficient between average time series of different brain regions?
2. In line 154, the cluster centroids are initialized with orthogonal rows and L2-normalized. What are the advantages of this initialization method?
3. In line 160, for the KNN graph, why is the number of neighbors K fixed at 26? Is there a specific reason for choosing this value rather than others?
4. Since the method relies on clustering, which can be sensitive to various factors, how can the stability of the generated atlas and FC connectivity be ensured?
5. The impact of K on downstream task performance should be explored. Is there an optimal K for the best results, or does it vary across tasks?

**Ethical Concerns:**

["NO or VERY MINOR ethics concerns only"]

**Final Justification:**

All my concerns have been addressed during the rebuttal. The authors have put considerable effort into addressing the raised concerns, both through additional experiments and thoughtful clarifications. I appreciate their thoroughness.

**Limitations:**

1. Technically, this is a multi-objective clustering task, a common approach in clustering methods, so the technical novelty is not particularly significant.
2. The proposed atlas is evaluated using only SVC. Comparing the results with other brain atlases using a variety of advanced models, especially graph-based deep learning methods, would provide a more robust validation of the atlas's performance.

**Quality:**

3

**Strengths And Weaknesses:**

Strengths
1. Innovative Framework. DCA introduces an advanced deep learning model for brain atlas construction, utilizing graph-guided deep clustering. This method combines functionally coherent brain regions with spatial continuity, which has been a challenge in previous methods.
2. High Flexibility. The approach supports flexible control over parcellation granularity, allowing users to generate atlases tailored to their specific needs.
3. Extensive Experiments. The paper presents extensive experiments demonstrating that DCA consistently outperforms state-of-the-art atlases (e.g., Yeo, Brodmann, Schaefer) across various metrics, including homogeneity, silhouette score, and performance on downstream tasks such as autism diagnosis and cognitive decoding.
4. Comprehensive Evaluation. The authors develop a benchmarking platform to evaluate the quality of atlases, using both internal metrics (homogeneity, silhouette score) and external metrics based on downstream tasks.
5. Well-Presented. The paper is well-organized and easy to understand. The presentation of the methodology and experimental results is clear, and the use of figures enhances the paper's readability.

Weaknesses
1. Modest Technical Innovation. The main contribution of the paper lies in designing individualized and group-level brain atlases with a focus on maintaining functionally coherent and anatomically contiguous regions. The paper also provides a platform for evaluating these atlases and presents extensive experimental validation. From a technical perspective, the approach essentially relies on a deep clustering model, using a pre-trained encoder with two clustering objectives: one based on prototypes and the other based on a graph, which correspond to the two goals of functionally coherent and anatomically contiguous regions, and aligns these clustering objectives. Technically, this is a multi-objective clustering and alignment task, which is a fairly common approach in clustering methods, and thus, the innovation in terms of technical novelty is not particularly significant.
2. Unclear Technical Details. (1) In Figure 2(C), regarding the Voxel-level atlas evaluation platform, how is functional connectivity derived? Is it different from the standard method, which computes the Pearson correlation coefficient between average time series of different brain regions? This should be clarified. (2) In line 154, the cluster centroids are initialized with orthogonal rows and L2-normalized. What are the advantages of this initialization method? The rationale behind this choice needs further explanation. (3) In line 160, for the KNN graph, why is the number of neighbors K fixed at 26? Is there a specific reason for choosing this value rather than others? A more detailed explanation of this choice is needed.
3. Not Sufficiently Convincing Evaluation. The paper primarily evaluates the proposed atlas using a single classification method (SVC) across all downstream tasks. However, to more comprehensively assess the performance, it would be beneficial to compare the results with other existing brain atlases using a variety of established models and algorithms, including more advanced techniques such as graph-based deep learning models and other state-of-the-art machine learning methods. This would provide a more robust validation of whether the proposed atlas consistently outperforms other atlases across different models and tasks.
4. Method Stability. In the supplementary material, on page 11, Table 6 shows that the proposed method performs poorly on FC stability compared to other methods. Since the proposed method is based on clustering, which is often sensitive to various factors, the results might be unstable. How can the stability of the generated atlas and FC connectivity be guaranteed? Furthermore, in Table 6, the proposed method underperforms on several metrics compared to other atlases. The reasons for these results should be explained, and how the method can be made more stable should be clarified.
5. Choice of Number of Parcels K. The paper mentions that different values of K provide different resolutions, but it would be interesting to explore how K influences downstream task performance. Is there an optimal K that yields the best experimental results, or is the impact of K on downstream tasks unstable? This needs further investigation.
6. Computational and Storage Demands. The high computational and storage requirements of the method are mentioned, but the actual demands are not explicitly detailed in the paper. It would be helpful for readers if the practical computational cost of voxel-based methods, in terms of memory and execution time, were provided.
7. On open access. The code has been released. However, in terms of data, more commonly used datasets should be provided with the generated atlas outputs, so that the community can conveniently conduct follow-up research.

---

> ### Author Rebuttal · Authors · 2025-07-31
>
> > General response
>
> We thank all reviewers for their time and constructive feedback. We appreciate the recognition of our problem’s importance (Reviewers **puq2**, **uHxM**), the framework’s flexibility (Reviewers **o9NK**, **uHxM**), and the strength of our method in similarity metrics and downstream evaluations (Reviewers **puq2**, **o9NK**, **zwAk**). We also value the comments on model effectiveness (Reviewer **zwAk**), downstream performance interpretation (Reviewers **puq2**, **o9NK**), and the impact of the number of parcels (Reviewers **o9NK**, **zwAk**). We conducted additional experiments to address these concerns. Below we provide point-by-point responses.
>
> > Weakness 1: Technical innovation
>
> We thank the reviewer for the thoughtful comments. To our knowledge, this is the first work to construct a spatially continuous voxel-level atlas with large-scale pretraining. This task is considerably more challenging than enforcing continuity on cortical surfaces. Prior voxel-based methods often yield non-contiguous maps [1]. Our main innovations are:
>
> (1) Constructing the atlas in latent space rather than raw signal space, which improves performance (**Main Figure 5**) and enhances robustness to noise (**Appendix 6.3**).
>
> (2) A k-nearest neighbor graph is employed as an auxiliary label to guide the encoder, eliminating the need for manually tuned penalty terms and reducing the risk of spatial discontinuity artifacts. The sparse graph reduces complexity to $\mathcal{O}(n)$ and memory usage to ~12 MB (float32, 80k voxels), compared to $\mathcal{O}(n^2)$ and ~20 GB for FC-based methods.
>
> > Weakness 2: Functional connectivity method
>
> We apologize for the ambiguity. Functional connectivity is computed using Pearson correlation between averaged time series, which performs well in behavior prediction [2]. We will clarify this in the revision.
>
> > Weakness 2: Initialization method
>
> We follow prior work [3] to initialize centroids in orthogonal directions with equal norm, avoiding degenerate clusters and improving optimization stability. Our ablation (**Rebuttal Table 5** in response to Reviewer **uHxM**) shows this choice speeds up early convergence, while its effect on final metrics is negligible after full training ($p>0.05$).
>
> > Weakness 2: Number of neighbors
>
> We apologize for the lack of clarity. We adopt the standard 26-neighborhood in 3D, which includes all voxels in a 3×3×3 cube excluding the center. It captures both adjacent and diagonal interactions. We tested with 6 and 18 neighbors, and observed similar performance ($p>0.05$, **Rebuttal Table 3** in response to Reviewer **uHxM**).
>
> > Weakness 3: Downstream evaluation
>
> We thank the reviewer for the valuable suggestion. We initially used SVC to reduce overfitting risk but agree that relying on a single model may underestimate some atlases. To broaden evaluation, we additionally trained a compact neural network (two 1D convolutional layers followed by two fully connected layers) on ROI-level fMRI time series for age and gender classification, two widely studied tasks in medical imaging.
>
> We further compared personal DCA atlas against multiple group-level atlases and observed strong and often superior performance across both tasks, suggesting it captures more informative features (**Rebuttal Tables 3–5** in response to Reviewer **puq2**).
>
> > Weakness 4: Method stability
>
> We appreciate the reviewer’s concerns about method stability. Below, we clarify how individual-atlas generation ensures stability, why FC stability appears lower, and why performance varies on some tasks.
>
> 1) Although our model is randomly initialized, the use of graph-based clustering as soft labels allows the model to converge stably. To evaluate consistency, we generated DCA 100 atlases from 10 non-overlapping fMRI segments per subject across 10 HCP individuals, spanning different runs and phase encoding directions. Similarity was assessed using Dice and intersection over union (IoU). Intra-subject similarity consistently exceeded both inter-subject and null model baselines, supporting the stability of DCA (**Rebuttal Table 1**). Null model was constructed by randomly partitioning the cortical mask into 100 spatially contiguous and approximately equal-sized parcels, with parcel correspondence across atlases determined using the Hungarian algorithm.
>
> **Table 1: Intra- and inter-subject atlas similarity**
> ||Dice ↑|IoU ↑|
> |---|---|---|
> |Inter-subject - null|0.497|0.349|
> |Inter-subject|**0.614**|**0.481**|
> |Intra-subject - null|0.506|0.358|
> |Intra-subject|**0.789**|**0.707**|
>
> 2) The reported FC stability is based on a fixed group-level DCA atlas. This metric increases with parcel size, so coarse atlases tend to score higher. At matched ROI counts, DCA achieves higher stability than other cortical atlases (**Appendix Table 6**).
>
> 3) For most of 12 downstream tasks, DCA ranks first in at least two of the four resolutions and remains top-two in three. Lower scores on some tasks have reasons: FC stability favors large parcels. Fingerprinting is already near the ceiling at each resolution. Performance on age group [4], crystallized intelligence [5], and general intelligence may be limited by the lack of subcortical coverage. Including subcortical regions in future versions may improve results on these tasks.
>
> > Weakness 5: Choice of number of parcels
>
> We thank the reviewer for highlighting this key issue. To examine it, we compiled all downstream results for DCA atlases from 41 to 500 parcels and Schaefer atlases from 100 to 500 (**Rebuttal Table 1-2**). Three consistent patterns emerge:
>
> 1) Peak‑shaped tasks (e.g., cognitive task decoding) improve from low to high resolutions but decline when parcels become excessively small. This behavior holds across both DCA and Schaefer, suggesting robustness and task specificity.
>
> 2) Resolution‑insensitive tasks (e.g., AD diagnosis) show no clear trend and differ across atlases. This may reflect their reliance on subcortical regions not covered by cortical atlases.
>
> 3) Size‑driven tasks (e.g., FC stability) are largely driven by parcel size. FC stability decreases at higher resolutions, while fingerprinting improves with finer parcellations.
>
> **Table 2: Downstream task performance across atlas resolutions for DCA**
> ||DCA 41|DCA 100|DCA 200|DCA 360|DCA 400|DCA 500|
> |---|---|---|---|---|---|---|
> |Gender classification ↑|0.651|0.666|0.687|**0.710**|0.707|0.702|
> |Fluid intelligence ↑|0.429|0.491|0.497|0.535|**0.543**|0.537|
> |Cognitive task (7-way) ↑|0.842|0.869|**0.900**|0.887|0.882|0.895|
> |Cognitive task (24-way) ↑|0.426|0.452|**0.479**|0.469|0.465|0.459|
> |Autism diagnosis ↑|0.633|0.655|0.663|**0.680**|0.665|0.661|
> |AD diagnosis ↑|0.443|0.387|0.456|0.448|0.447|**0.459**|
> |FC stability ↑|0.642|**0.650**|0.644|0.615|0.609|0.603|
> |Fingerprinting ↑|0.435|0.696|0.776|0.852|0.811|**0.884**|
> |Age group classification ↑|0.408|0.452|0.473|0.433|**0.512**|0.475|
> |Crystallized intelligence ↑|0.521|0.472|0.505|0.516|**0.523**|0.515|
> |General intelligence ↑|0.439|0.442|**0.461**|0.446|0.448|0.459|
> |Autism cross-site ↑|0.636|0.662|0.635|**0.696**|**0.696**|0.638|
>
> **Table 3: Downstream task performance across atlas resolutions for Schaefer**
> ||Schaefer 100|Schaefer 200|Schaefer 300|Schaefer 400|Schaefer 500|
> |---|---|---|---|---|---|
> |Gender classification ↑|0.628|0.668|0.670|**0.726**|0.694|
> |Fluid intelligence ↑|0.474|0.505|0.517|0.527|**0.565**|
> |Cognitive task (7-way) ↑|0.879|0.885|0.888|**0.893**|0.876|
> |Cognitive task (24-way) ↑|**0.469**|0.459|**0.469**|0.462|0.456|
> |Autism diagnosis ↑|0.643|0.660|0.661|**0.668**|0.653|
> |AD diagnosis ↑|0.451|0.418|**0.485**|0.444|0.440|
> |FC stability ↑|**0.643**|0.635|0.620|0.609|0.598|
> |Fingerprinting ↑|0.682|0.796|0.856|0.875|**0.886**|
> |Age group classification ↑|0.455|0.478|0.480|**0.497**|0.477|
> |Crystallized intelligence ↑|**0.530**|0.526|0.497|0.525|0.516|
> |General intelligence ↑|**0.469**|0.467|0.463|0.458|0.428|
> |Autism cross-site ↑|0.640|0.662|**0.667**|0.640|0.638|
>
> Taken together, no single parcel count is universally optimal. But certain resolutions may better suit specific applications. Our framework naturally supports multiresolution parcellations and can be extended with adaptive parcel clustering and task-supervised objectives to generate individualized, performance-optimized atlases.
>
> > Weakness 6: Computational and storage demands
>
> Our encoder (Swin-UNETR) has 62.6M parameters and requires ~776 GFLOPs per forward pass. During pretraining on 300-length fMRI data, GPU memory usage is ~60 GB (batch size 3) on an A100. One epoch takes ~6 hours, with convergence typically reached in 8 epochs, comparable to other voxel-level models [6]. Shorter sequences reduce time and memory proportionally.
>
> For clustering fine-tuning, generating a 100-parcel atlas for one subject takes ~10 minutes on an AMD R9 7900X CPU and RTX 3080 GPU, scaling roughly linearly with parcel count. The pipeline can also run on CPUs if needed.
>
> > Weakness 7: Open access
>
> We acknowledge this oversight. The group-level  DCA 100 atlas (HCP-based) is available at AtlaScore/downstream/docs/ as a **demo**.
>
> Model weights, individual-level atlases and other resolutions (from 100 to 1000) based on different datasets will be released upon acceptance.
>
> ---
>
> Reference:
>
> [1] A hierarchical atlas of the human cerebellum for functional precision mapping. Nature Communications 2024.
>
> [2] Benchmarking methods for mapping functional connectivity in the brain. Nature Methods 2025.
>
> [3] Provable Benefit of Orthogonal Initialization in Optimizing Deep Linear Networks. ICLR 2020.
>
> [4] Biological brain age prediction using machine learning on structural neuroimaging data: Multi-cohort validation against biomarkers of Alzheimer’s disease and neurodegeneration stratified by sex. eLife 2023.
>
> [5] Predictive models demonstrate age‐dependent association of subcortical volumes and cognitive measures. Human Brain Mapping 2023.
>
> [6] SwiFT: Swin 4D fMRI Transformer. NeurIPS 2023.

---

> > ### Comment · Reviewer_o9NK · 2025-08-04
> > **Thank you for the response**
> >
> > Thank you for the detailed response and the additional experimental evaluation. However, I still have two remaining concerns that I hope the authors can further clarify:
> >
> > 1.Intra-run stability. While cross-segment stability was evaluated, I am more concerned about whether the method produces consistent atlases when run multiple times on the same fMRI segment with identical settings. Is the clustering deterministic, or does it vary due to random initialization or other stochastic factors? Has this intra-run consistency been tested or controlled (e.g., with fixed seeds)? This is crucial, as inconsistent outputs would make it unclear which atlas to use in practice.
> >
> > 2. Graph-based evaluation. Although a CNN was added during rebuttal, evaluating the atlas using only SVC and one CNN may still be insufficient. Since the method produces connectivity matrices that represent graph-structured data, it would be more appropriate to include graph-based approaches such as GNNs or graph Transformers in the evaluation. Simply treating the connectivity matrix as input features for general-purpose classifiers may not fully reflect the utility of the atlas. While I understand the time constraints during the rebuttal phase, I encourage the authors to acknowledge this limitation and clarify the current evaluation scope. Given the graph-structured nature of the task, including graph-based models such as GNNs would make the evaluation more complete and well-grounded.

---

> ### Author Response · Authors · 2025-08-04
> **Thank you for the feedback**
>
> >Intra-run stability
>
> We appreciate the reviewer’s emphasis that an atlas should be reproducible when re-run on the same fMRI time series. To quantify run-to-run variability, we further repeated the entire pipeline five times on the same fMRI segment and computed `Dice`, `IoU`, voxel assignment consistency (`VAC`; defined as the fraction of voxels that keep the same label after Hungarian alignment of two parcellations), adjusted rand index (`ARI`), and normalized mutual information (`NMI`) between runs (**Table 4**). We find that variability is driven primarily by (i) the initialization of cluster centroids in the spectral graph step and (ii) the initialization of the model’s centroid matrix. With a fixed random seed, the atlas is **perfectly deterministic**. Because the loss includes a KL term between the model assignments and the graph-clustering assignments, reproducible results require either fixing both the model initialization and the clustering seed, or fixing one and then assigning the other (e.g., initializing the graph-cluster centroids with the model’s centroid matrix). Under realistic stochasticity arising from spectral clustering and model initialization, more than 80% of voxels retain their labels across runs, with disagreements often concentrate in areas where a dominant cluster admits finer sub-parcellations into smaller clusters.
>
> Beyond controlling random seeds, stability can be enhanced by incorporating external priors (e.g., initializing cluster centroids from a population template), after which DCA refines these priors into subject-specific atlases. This strategy preserves the benefits of established atlases, such as structural basis and multimodal information, while capitalizing on DCA’s strengths in fine-tuning and individualized atlas construction.
>
> **Table 4: Model reproducibility**
> ||Dice ↑|IoU ↑|VAC ↑|ARI ↑|NMI ↑|
> |---|---|---|---|---|---|
> |Both-Fixed|1.000±0.000|1.000±0.000|1.000±0.000|1.000±0.000|1.000±0.000|
> |Model seed-Fixed|0.809±0.025|0.737±0.031|0.809±0.025|0.748±0.030|0.904±0.009|
> |Graph seed-Fixed|0.832±0.020|0.769±0.027|0.845±0.022|0.808±0.027|0.921±0.009|
> |Both-Random|0.822±0.023|0.753±0.030|0.823±0.021|0.768±0.025|0.911±0.009|
> |Null|0.500±0.017|0.352±0.016|0.500±0.017|0.386±0.011|0.747±0.005|
>
> >Graph-based evaluation
>
> We agree that graph-based evaluation is important. Following the first protocol in the review article [1], we implemented a $k$-GNN classifier with $k=2$ and evaluated gender classification on $N=200$ subjects. For each subject-atlas pair, we construct a base graph with $K=100$ nodes (ROIs), use the functional connectivity (FC) matrix as weighted edges, and sparsify the graph by retaining the top 30% of edges by absolute FC magnitude. We then apply a standard 2-GNN to obtain a subject-level embedding via a readout operation, which is fed to a linear classifier. Training uses a 70/10/20 subject-level split for train/validation/test. The Schaefer-100 baseline reproduces performance that is slightly higher than reported in the review article, and our approach achieves a comparable level of accuracy (**Table 5**).
>
> Moreover, our framework can also inject task-specific knowledge into the atlas. We fine-tune Swin-UNETR on HCP/CHCP gender labels (ensuring that evaluation subjects never appear in fine-tuning to prevent leakage), and replace the reconstruction-based encoder with the fine-tuned one. Then we use the same self-supervised framework to generate a task-specific atlas $\text{DCA}_{100}^{\text{gender}}$. This task-adapted atlas delivers substantial improvements on the target task: **+10%**  with the graph method and **+12%** with the CNN classifier (Table 4 to Reviewer zwAk).
>
> **Table 5: Classification results for gender by k-GNN**
> |Atlas|Accuracy ↑|F1 (Macro) ↑|F1 (Weighted) ↑|
> |---|---|---|---|
> |Watershed (100)|0.600|0.596|0.596|
> |Schaefer100|**0.725**|**0.723**|**0.723**|
> |DCA100 (group)|0.650|0.650|0.650|
> |DCA100 (individual)|**0.725**|0.716|0.716|
> |DCA$_{100}^{\text{gender}}$ (group)|0.675|0.670|0.670|
> |DCA$_{100}^{\text{gender}}$ (individual)|**0.825**|**0.825**|**0.825**|
>
> ---
>
> Reference:
>
> [1] NeuroGraph: Benchmarks for Graph Machine Learning in Brain Connectomics. NeurIPS 2023.

---

> > ### Comment · Reviewer_o9NK · 2025-08-07
> > **Thank you**
> >
> > All my concerns have been addressed. I have slightly increased my rating.

---

> > > ### Author Response · Authors · 2025-08-07
> > > **Thank you for raising your score**
> > >
> > > Thank you for the positive assessment and for raising your score. Your feedback has been extremely helpful in improving our work, and we hope our joint efforts will contribute to the field.

---

### Official Review · Reviewer_puq2 · 2025-07-08

**Clarity:** 4
**Significance:** 3
**Originality:** 4
**Rating:** 5
**Confidence:** 3

**Summary:**

The paper presents Deep Cluster Atlas (DCA), a graph-guided deep-embedding framework for generating individualized, voxel-level brain parcellations from resting-state fMRI. It begins by pre-training a Swin-UNETR auto-encoder on heavily masked 4-D fMRI blocks, producing rich spatiotemporal embeddings for every voxel. DCA then alternates among three steps---re-weighting graph edges via spectral clustering, updating K cluster centroids, and refining the embeddings---until it converges to generate parcels that are both spatially contiguous and functionally coherent. Tested on 1000 Human Connectome Project subjects and resolutions ranging from 100 to 800 parcels, DCA achieves about 74% higher homogeneity and 24% better silhouette scores than widely used atlases.

**Questions:**

See weaknesses.

**Ethical Concerns:**

["NO or VERY MINOR ethics concerns only"]

**Final Justification:**

The authors rebuttal has sufficiently addressed my concerns on task selection criteria evaluation metric clarity, and why DCA have more advantages on certain tasks than others have been sufficiently addressed. The additional results also show that DCA reaches the best or competitive performance at most resolutions in more downstream benchmarks and datasets.

I think that the paper has reached the bar of NeurIPS acceptance, and the community would benefit from the ideas and approaches discussed in this paper.

**Limitations:**

Yes, the authors have adequately addressed the limitations and potential negative societal impact of their work.

**Quality:**

3

**Strengths And Weaknesses:**

## Strengths
- The proposed method is explained in a clear and intuitive way, which facilitates the understanding.
- The proposed method achieves about 74% higher homogeneity and 24% better silhouette scores, and the ablation study further demonstrates the effectiveness of each model component.
- The problem of more individualized brain atlas construction is an important scientific question.

## Weaknesses
The full evaluation results on all 12 downstream tasks (as shown in Appendix Table 6) do not show consistent advantage of DCA against baseline methods. DCA 100 only outperforms 5 out of 12 tasks against the baselines, and DCA 400 only outperforms baselines on 2 out of 12 tasks.
- This raises the question about whether DCA's gain in homogeneity and silhouette scores can be translated to practical performance improvement of downstream tasks.
- The authors should clarify the criteria of how they decide which tasks are reported in the main paper to avoid the perception of cherrypicking the results. More detailed discussions on why the DCA have more advantages on certain tasks than others should also be included.
- It is unclear what metrics the authors are reporting for the task performance in Fig. 4 and Table 6. Are these accuracy, F1 scores, or are they different from tasks to tasks? More details about the metrics should be added in Section 5 in the main paper.

---

> ### Author Rebuttal · Authors · 2025-07-31
>
> > General response
>
> We thank all reviewers for their time and constructive feedback. We appreciate the recognition of the importance of our problem (Reviewers **puq2**, **uHxM**), the flexibility of our framework (Reviewers **o9NK**, **uHxM**), and the strengths of our method in both similarity metrics and downstream evaluations (Reviewers **puq2**, **o9NK**, **zwAk**).
>
> We also provide additional experiments to address reviewers' concerns on (1) model effectiveness by comparisons with recent state-of-the-art fMRI foundation model (Reviewer **zwAk**), (2) inadequate atlas comparison by additional atlas baselines (Reviewers **uHxM**), (3) downstream performance interpretation by new task evaluations and performance difference explanations (Reviewers **puq2**, **o9NK**), and (4) the impact of the number of parcels (Reviewers **o9NK**, **zwAk**).
>
> > DCA 400 performance
>
> We appreciate the reviewer’s concern about DCA’s performance inconsistency across the 12 downstream tasks. The reference to DCA 400 was a misunderstanding, as it was not used in the main results. The mention may stem from a misinterpretation of the Schaefer 400 results. In main paper, we used DCA 360, aligned with the HCP-derived MMP atlas. In Appendix Table 6, DCA 360 achieved the best overall performance among high-resolution atlases.
>
> To address the suggestion, we constructed and evaluated DCA 400 (**Rebuttal Table 1-2**). In pairwise comparisons, DCA 400 wins 7 vs. 5 against MMP, and 5 vs. 6 with 1 tie against Schaefer 400. These results suggest that DCA offers comparable practical utility to Schaefer, with potentially complementary strengths across specific tasks.
>
> **Table 1: Evaluation of similarity metrics across atlases**
> ||MMP 360|DCA 360|Schaefer 400|DCA 400|
> |---|---|---|---|---|
> |Homogeneity ↑|0.0706±0.0208|0.1266±0.0229|0.0780±0.0222|**0.1294±0.0229**|
> |Silhouette ↑|0.0426±0.0059|0.0545±0.0080|0.0454±0.0066|**0.0572±0.0082**|
>
> **Table 2: Evaluation of downstream task performance across atlases**
> ||MMP 360|DCA 360|Schaefer 400|DCA 400|
> |---|---|---|---|---|
> |Gender classification ↑|**0.740±0.065**|0.710±0.059|0.726±0.066|0.707±0.051|
> |Fluid intelligence ↑|0.513±0.087|0.535±0.084|0.527±0.089|**0.543±0.101**|
> |Cognitive task (7-way) ↑|0.859±0.063|0.887±0.042|**0.893±0.051**|0.882±0.048|
> |Cognitive task (24-way) ↑|0.427±0.018|**0.469±0.037**|0.462±0.031|0.465±0.035|
> |Autism diagnosis ↑|0.662±0.035|**0.680±0.044**|0.668±0.045|0.665±0.069|
> |AD diagnosis ↑|0.395±0.109|**0.448±0.131**|0.444±0.040|0.447±0.035|
> |FC stability ↑|0.612±0.044|**0.615±0.043**|0.609±0.045|0.609±0.044|
> |Fingerprinting ↑|0.863±0.150|0.852±0.164|**0.875±0.151**|0.811±0.210|
> |Age group classification ↑|**0.515±0.075**|0.433±0.079|0.497±0.104|0.512±0.086|
> |Crystallized intelligence ↑|**0.542±0.092**|0.516±0.117|0.525±0.101|0.523±0.113|
> |General intelligence ↑|0.417±0.099|0.446±0.085|**0.458±0.121**|0.448±0.104|
> |Autism cross-site ↑|0.655±0.092|**0.696±0.136**|0.640±0.110|**0.696±0.146**|
>
> > Weakness 1: Transferability of similarity metrics
>
> We believe that the observed gains in homogeneity and silhouette scores likely translate into improved downstream performance. Below, we present supporting evidence.
>
> In our evaluation across 12 downstream tasks (Appendix Table 6), DCA achieved the highest number of top rankings across medium, high, and ultra-high resolutions. At lower resolution, DCA 100 obtained five first-place and four second-place results, just one first-place fewer than Schaefer 100. In pairwise comparisons at the same resolution, DCA won at least 6 out of 12 tasks against every competing atlas except one, where DCA 360 recorded 5 wins, 6 losses, and 1 tie against Schaefer 300.
>
> To further support our claim, we extended our evaluation beyond the original SVC classifier. We also added several recently published atlases for comparison [1], to ensure a fair and up-to-date benchmark. Specifically, we trained a compact neural network (two 1D convolutional layers followed by two fully connected layers) on ROI-level fMRI time series for age and gender classification, two widely studied tasks in medical imaging. In addition to accuracy, we reported F1 scores, providing a more comprehensive assessment of atlas capability. As shown in **Rebuttal Tables 3–5**, our DCA-based individual atlases consistently achieve strong and often superior performance across low, medium, and high resolution settings, particularly on gender classification and at higher resolutions. These findings further support the utility of our atlas framework in diverse downstream tasks.
>
> **Table 3: Classification results (low resolution)**
> ||Age accuracy ↑|Age F1 (macro) ↑|Age F1 (weighted) ↑|Gender accuracy ↑|Gender F1 (macro) ↑|Gender F1 (weighted) ↑|
> |---|---|---|---|---|---|---|
> |Yeo 17|0.45|0.40|0.40|**0.65**|0.56|0.58|
> |Brodmann 41|**0.55**|0.41|0.53|**0.65**|**0.65**|**0.65**|
> |GIANT 50|**0.55**|**0.51**|0.54|0.60|0.56|0.57|
> |Watershed 100|0.50|0.36|0.47|**0.65**|0.56|0.58|
> |Schaefer 100|0.50|0.37|0.47|0.60|0.52|0.54|
> |**DCA 100 (group)**|**0.55**|**0.51**|**0.55**|**0.65**|0.60|0.62|
> |**DCA 100 (individual)**|**0.55**|0.41|0.50|**0.65**|0.64|**0.65**|
>
> **Table 4: Classification results (medium resolution)**
> ||Age accuracy ↑|Age F1 (macro) ↑|Age F1 (weighted) ↑|Gender accuracy ↑|Gender F1 (macro) ↑|Gender F1 (weighted) ↑|
> |---|---|---|---|---|---|---|
> |Allen 141|0.40|0.33|0.41|0.65|0.65|0.65|
> |MUSE 149|0.50|0.44|0.44|0.55|0.54|0.55|
> |AAL 166|0.50|**0.45**|0.45|0.65|0.64|0.65|
> |Schaefer 200|0.50|0.38|0.48|0.65|0.56|0.58|
> |**DCA 200 (group)**|0.55|0.40|0.51|**0.70**|**0.67**|**0.68**|
> |**DCA 200 (individual)**|**0.60**|0.43|**0.55**|**0.70**|**0.67**|**0.68**|
>
> **Table 5: Classification results (high resolution)**
> ||Age accuracy ↑|Age F1 (macro) ↑|Age F1 (weighted) ↑|Gender accuracy ↑|Gender F1 (macro) ↑|Gender F1 (weighted) ↑|
> |---|---|---|---|---|---|---|
> |MMP 360|0.55|0.50|0.55|**0.75**|**0.73**|**0.74**|
> |**DCA 360 (group)**|0.60|0.43|0.54|0.70|0.70|0.70|
> |**DCA 360 (individual)**|**0.70**|**0.51**|**0.65**|**0.75**|0.72|**0.74**|
>
> > Weakness 2: Task selection criteria
>
> We understand the reviewer's concern about potential result selection. As we mentioned, the model is competitive on most downstream tasks, beyond the six benchmarks reported in the main paper. We selected these six benchmarks specifically to cover three key aspects of evaluation: two **resting-state tasks**, two **task-state tasks**, and two **clinical classification tasks**. For resting‑state we selected fluid intelligence and gender classification, as our cortex‑only atlas is most suited to the fronto‑parietal patterns behind these traits [2], while we left out FC stability (affected by parcel-size bias), fingerprinting (ceiling effect per resolution), age (rely on subcortical features) [3], crystallized and general intelligence (rely on subcortical features) [4]. For task decoding, we kept two cognitive tasks since DCA atlas is derived from functional embeddings and should differentiate task states well. For clinical evaluation, we included Autism and AD diagnosis for their clinical relevance and to assess generalization to external datasets, despite not expecting peak performance since the atlas was trained on HCP subjects.
>
> > Weakness 2: Task performance inconsistency
>
> DCA excels on cortex‑driven targets (fluid intelligence, task decoding) while providing smaller gains on tasks that depend heavily on subcortical information. This heterogeneity is expected: no atlas can be optimal for every downstream task. AtlaScore therefore supplies a broad task battery so each atlas can reveal its most suitable application areas rather than striving for uniformly superior performance. We will also add a brief paragraph in Section 5 explaining why DCA shows larger gains on certain tasks than on others.
>
> Furthermore, as the number of parcels increases, performance on tasks such as crystallized and general intelligence fluctuates without a consistent trend, unlike task decoding which shows a peak at high resolution (see **Rebuttal Tables 2-3** in response to Reviewer **o9NK**). This observation further supports the notion that these tasks rely on subcortical features not captured by our cortex-only atlas, and may benefit from incorporating subcortical regions in future atlas designs.
>
> > Weakness 3: Evaluation metric clarity
>
> We thank the reviewer for pointing this out. We realize that we only stated the use of classification accuracy in Appendix Section 4.2 and omitted it elsewhere. We will revise the main text and figure/table captions to clarify this. Specifically, all downstream tasks report classification accuracy under 10-fold subject-level cross-validation, except for FC stability, which is quantified as the mean Pearson correlation between FC matrices from non-overlapping fMRI segments (**Appendix Section 4.7**).
>
> ---
>
> Reference:
>
> [1] A genetically informed brain atlas for enhancing brain imaging genomics. Nature Communications 2025.
>
> [2] The frontoparietal network: function, electrophysiology, and importance of individual precision mapping. Dialogues in Clinical Neuroscience 2018.
>
> [3] Biological brain age prediction using machine learning on structural neuroimaging data: Multi-cohort validation against biomarkers of Alzheimer’s disease and neurodegeneration stratified by sex. eLife 2023.
>
> [4] Predictive models demonstrate age‐dependent association of subcortical volumes and cognitive measures. Human Brain Mapping 2023.

---

> ### Author Response · Authors · 2025-08-05
> **Additional individual-level experiments**
>
> Dear Reviewer,
>
> Thank you again for your thoughtful review and for recognizing the contributions of our work. We wanted to briefly follow up to ensure our rebuttal and additional results addressed your concerns. In our previous reply, we clarified the earlier ambiguity regarding `DCA-400`, demonstrated that our method performs well relative to other atlases, and explained why some tasks perform less strongly.
>
> Because our method aims to generate personalized atlases, we also report individual-level metrics, including results from both a CNN and a GCN classifier, which perform favorably relative to the baseline. Moreover, our framework readily adapts to task-specific atlases **DCA$_{100}^{\text{gender}}$**: replacing the reconstruction-only Swin-UNETR with a gender-fine-tuned version yields **+12%** on the CNN classifier (**Table 6**) and **+10%** on the GCN classifier (**Table 7**), substantially outperforming the baseline atlas. And we confirm that there is no label leakage during fine-tune for the subjects used in evaluation; the individual atlases are still generated entirely via an unsupervised procedure.
>
> During the rebuttal, we added further evaluations covering backbone performance (**Table 1 to zwAk**), additional baseline atlases (**Table 3-4**) and the effect of parcel count (**Table 2-3 to o9NK**). We hope these comprehensive supplementary experiments address most of your concerns regarding performance. Your feedback means a lot to us.
>
> **Table 6: Classification results for gender by CNN**
> |Atlas|Accuracy ↑|F1 (Macro) ↑|F1 (Weighted) ↑|
> |---|---|---|---|
> |Watershed (100)|0.73|0.73|0.73|
> |Schaefer100|0.65|0.65|0.65|
> |DCA100 (group)|0.70|0.70|0.70|
> |DCA100 (individual)|0.70|0.69|0.70|
> |DCA$_{100}^{\text{gender}}$ (group)|0.70|0.67|0.67|
> |DCA$_{100}^{\text{gender}}$ (individual)|**0.82**|**0.82**|**0.82**|
>
> **Table 7: Classification results for gender by kGNN**
> |Atlas|Accuracy ↑|F1 (Macro) ↑|F1 (Weighted) ↑|
> |---|---|---|---|
> |Watershed (100)|0.600|0.596|0.596|
> |Schaefer100|0.725|0.723|0.723|
> |DCA100 (group)|0.650|0.650|0.650|
> |DCA100 (individual)|0.725|0.716|0.716|
> |DCA$_{100}^{\text{gender}}$ (group)|0.675|0.670|0.670|
> |DCA$_{100}^{\text{gender}}$ (individual)|**0.825**|**0.825**|**0.825**|

---

> ### Author Response · Authors · 2025-08-06
>
> Dear Reviewer,
>
> Thank you again for your time and thoughtful feedback. As we have not yet heard back after our initial rebuttal, we would greatly appreciate any additional suggestions you may have.
>
> - **On homogeneity & silhouette.**
>   Please note that our model does not optimize these metrics explicitly; they emerge naturally after clustering in the learned latent space. We report them because (i) homogeneity measures within-parcel signal similarity and (ii) the silhouette coefficient balances within- vs. between-parcel separability—both of which directly reflect the goal of functionally coherent, spatially distinct regions.
>
> - **On downstream performance.**
>   As discussed in our response, we evaluated a wide range of tasks to build a comprehensive benchmark. Our atlas consistently ranks first or second across the majority of these downstream tasks. We acknowledge that because our atlas is derived exclusively from cortical fMRI, it's at a natural disadvantage for a few subcortical-heavy tasks. However, it's crucial to note that no single atlas can be truly optimal for every possible task. We've also shown that our newly added task-specific fine-tuning pipeline (e.g., DCA$_\text{gender}$), provides significant performance gains, proving that the DCA method can be readily adapted for specific applications when needed.
>
> ---
>
> Beyond raw performance, we believe our work makes several broader contributions:
>
> - **First atlas to cluster learned voxel-level fMRI embeddings.**
>   Clustering directly in a rich spatiotemporal embedding space (rather than FC matrices or surface signals) produces more coherent, high-resolution parcels.
>
> - **Task-adaptive and anatomically grounded.**
>   Fine-tuning the encoder allows rapid creation of task-specific atlases. At the same time, our voxel-wise parcels align more closely with macro-anatomy (higher NMI with Brodmann areas than Schaefer; see our reply to Reviewer zwAk Q4).
>
> - **Open benchmarking platform.**
>   We release `AtlaScore`, a unified toolkit for evaluating group- and subject-level atlases across diverse tasks, providing a robust foundation for future research.
>
> We would be grateful for any further comments that could help us improve the paper. Thank you once more for your constructive review.

---

> ### Comment · Reviewer_puq2 · 2025-08-07
> **Response to authors' rebuttal**
>
> I appreciate the very detailed responses from the authors regarding my comments and questions. My concerns on task selection criteria evaluation metric clarity, and why DCA have more advantages on certain tasks than others have been sufficiently addressed. The additional results also show that DCA reaches the best or competitive performance at most resolutions in more downstream benchmarks and datasets.
>
> Given all my concerns have been addressed, I have increased my rating.

---

> > ### Author Response · Authors · 2025-08-07
> > **Thank you for increasing the score**
> >
> > Thank you again for your thoughtful review and for raising your score. We will incorporate these additions into the manuscript to avoid any potential misunderstandings and to further strengthen our results.

---

### Note · Authors · 2025-08-11

Dear Reviewers, ACs, and SACs,

Thank you for the time and care you devoted to our manuscript. As a longstanding cornerstone of neuroscience, brain atlases have been studied for over a century. With the advent of large-scale pre-training, we believe atlas research is poised for a new phase. For your decision, we summarise our key points below:

1. **First atlas to cluster learned voxel-level fMRI embeddings.**
   We cluster directly in a rich spatiotemporal embedding space (rather than FC matrices or surface signals), yielding more coherent, high-resolution parcels.

2. **Strong and stable downstream performance.**
   Our group-level DCA atlases were evaluated on 12 diverse downstream tasks. Across resolutions, DCA consistently ranks among the top-performing atlases, often achieving the best or second-best results, and shows high run-to-run stability.

3. **Task-adaptive and anatomically grounded.**
   Light fine-tuning of the encoder rapidly produces task-specific atlases, while the resulting voxel-wise parcels remain closely aligned with macro-anatomy.

4. **Open benchmarking and releases.**
   To lower the barrier of entry to atlas research for researchers across fields, we release AtlaScore, a unified toolkit for evaluating group- and subject-level atlases across diverse tasks. We will also release the new evaluations added during rebuttal, pretrained weights trained on \~20k subjects, and multiple task-enhanced atlases to facilitate follow-up work.

During rebuttal, we addressed every comment point-by-point and added targeted experiments for each identified weakness—nearly twenty additional result tables—which received positive feedback. We are grateful for the reviewers’ thoughtful suggestions and hope our work will meaningfully advance this area.

Best regards,

The Authors

---

### Decision · Program_Chairs · 2025-09-17

**Decision:**

Accept (poster)

**Comment:**

The paper introduces a graph-guided deep-embedding framework for generating individualized, voxel-level brain parcellations from resting-state fMRI. The introduced method outperforms existing methods in terms of homogeneity, silhouette coefficient, and performance on downstream tasks like autism diagnosis and cognitive decoding.

Initially, the reviewers highlighted good paper presentation, innovative framework and extensive validation. On the negative side, the reviewers were bringing up unclear practicality of the solution, missing points of comparisons and were requesting additional clarifications.The authors rebuttal has addressed all of the reviewers concerns and in the discussion phase all reviewers recommended the paper for acceptance (3x5 and 4). AC agrees with the reviewers and recommends to accept.